# How to measure and evaluate binding affinities

**Inga Jarmoskaite[1†]\*, Ishraq AlSadhan[1†], Pavanapuresan P Vaidyanathan[1†‡], Daniel Herschlag[1,2,3]\***

[1]Department of Biochemistry, Stanford University, Stanford, United States; [2]Department of Chemical Engineering, Stanford University, Stanford, United States; [3]Stanford ChEM-H, Stanford University, Stanford, United States

**Abstract** Quantitative measurements of biomolecule associations are central to biological understanding and are needed to build and test predictive and mechanistic models. Given the advances in high-throughput technologies and the projected increase in the availability of binding data, we found it especially timely to evaluate the current standards for performing and reporting binding measurements. A review of 100 studies revealed that in most cases essential controls for establishing the appropriate incubation time and concentration regime were not documented, making it impossible to determine measurement reliability. Moreover, several reported affinities could be concluded to be incorrect, thereby impacting biological interpretations. Given these challenges, we provide a framework for a broad range of researchers to evaluate, teach about, perform, and clearly document high-quality equilibrium binding measurements. We apply this framework and explain underlying fundamental concepts through experimental examples with the RNA-binding protein Puf4.

**\*For correspondence:**
ijarmosk@stanford.edu (IJ);
herschla@stanford.edu (DH)

[†]These authors contributed equally to this work

**Present address:** [‡]Protillion Biosciences, Burlingame, United States

**Competing interests:** The authors declare that no competing interests exist.

## Introduction

Molecular associations lie at the heart of biology. Their thermodynamics provides information critical for deriving a fundamental understanding of molecular functions. In a broader biological context, these associations are linked and interconnected in complex networks that allow sensitive and precise developmental programs and responses to environmental cues, and that are altered in disease states. The outputs of pathways and networks are determined by the quantitative interplay of their many constituent molecules and interactions. Thus, equilibrium constants for association between network components are needed to define, model, predict, and ultimately precisely manipulate biology.

A limitation of traditional biochemical measurements is their low throughput, especially in relation to the large number of cellular interactions. Excitingly, several strategies have recently emerged to obtain high-throughput, quantitative information for intermolecular associations (e.g. *Buenrostro et al., 2014*; *Tome et al., 2014*; *Lambert et al., 2014*; *Nutiu et al., 2011*; *Maerkl and Quake, 2007*; *Adams et al., 2016*; *Jain et al., 2017*). Given these potentially transformative advances, it is especially timely to assess the accuracy of equilibrium binding measurements. We wanted to know whether current practices are sufficient to ensure reliable and accurate measurements, and whether the reliability of these measurements can be readily ascertained from the information provided in published work.

Our survey of 100 literature binding measurements, presented below, uncovered recurring problems with a large majority of studies. Fortunately, there are straightforward procedures, laid out here, that can be followed to ensure that published binding measurements are reliable. The principles underlying these procedures have been discussed and we build on these previous reports (*Pollard, 2010*; *Hulme and Trevethick, 2010*; *Sanders, 2010*). We focus on a minimal set of critical

actionable steps and controls that biologists of any background should be able to implement in their binding measurements. We apply these procedures with experimental examples and also demonstrate the pitfalls of omitting essential controls. To further streamline application of these standard procedures, we provide a convenient checklist that can organize and guide experiments and can be used as an aid in summarizing and presenting results for publication.

## Results

### Assessing the current state of binding measurements

We evaluated published binding measurements using RNA-protein interactions as an illustrative example. We surveyed 100 studies that reported equilibrium dissociation constants ($K_D$ values) and

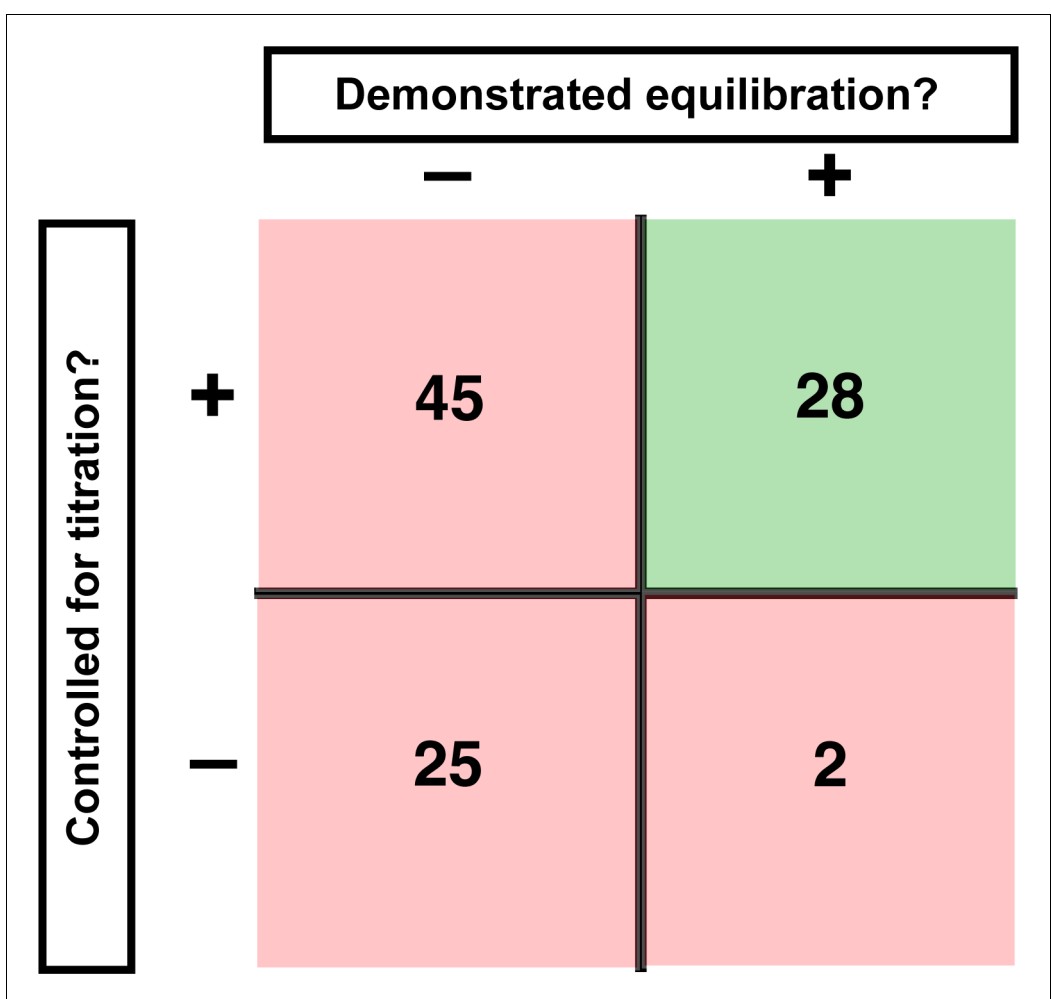

**Figure 1.** Assessment of published $K_D$ values for RNA-binding proteins. We analyzed 100 papers reporting $K_D$ or 'apparent $K_D$' values of RNA/protein interactions. Measurements were evaluated based on two criteria: demonstrating equilibration (horizontal axis) and controlling for titration (vertical axis). Detailed criteria are described in Materials and methods, and the source data are provided in *Supplementary file 1*. The right column includes predominantly studies that used ITC and SPR, techniques that inherently record binding progress over time (24/30 in this column). The fraction of studies that varied time to demonstrate equilibration in non-ITC/SPR experiments is considerably smaller (6 of the 76 papers that did not exclusively use ITC or SPR, or <10%). The online version of this article includes the following figure supplement(s) for figure 1:

**Figure supplement 1.** Survey of incubation times for published equilibrium dissociation constants.
**Figure supplement 2.** Survey of titration controls in published binding studies.

scored them based on two key criteria for reliable binding measurements: sufficient time to equilibration and proper concentration regime (*Figure 1*).

First, we asked if equilibration was demonstrated. By definition, an equilibrium state is invariant with time. So, determining a binding equilibrium constant requires showing that there is no change in the amount of bound complex over time. Of the 100 studies surveyed, 70 did not report varying time for reported equilibrium measurements (*Figure 1*; *Supplementary file 1*). Of the 30 studies that did vary time, 24 exclusively used techniques with built-in monitoring of progress over time (isothermal titration calorimetry (ITC) and surface plasmon resonance [SPR]). Of the remaining 76 studies—those using approaches such as native gel shifts, nitrocellulose filter binding, and fluorescence anisotropy—less than 10% reported varying time (*Figure 1*, *Figure 1—figure supplement 1*).

We know from individual discussions that some researchers carry out these controls, as we advocate below, but do not report them. Unfortunately, the published record then cannot distinguish between these studies and others that have not demonstrated equilibration.

A second critical control entails demonstrating that the $K_D$ is not affected by titration, as artifacts can arise when the concentration of the constant limiting component is too high relative to the dissociation constant ($K_D$). Similar to varying time to establish equilibrium, systematically varying the concentration of the limiting component provides a definitive control for effects of titration. In our survey, only 5% of studies reported performing this or equivalent control (*Figure 1—figure supplement 2*). Nevertheless, most authors appeared to be aware of the need to avoid titration, as the majority of studies (~70%) reported using appropriately low concentrations of the limiting component or employed advanced analysis methods. We consider these examples as reasonably titration-controlled for the purpose of the survey, but emphasize the importance of empirical controls in the sections below. Importantly, this leaves, at a minimum, one-fourth of studies at risk for titration (*Figure 1*, *Figure 1—figure supplement 2*).

To what extent do these limitations affect the reported equilibrium binding constants in practice? As an example, for Puf4 binding (see below), *not* controlling for the factors above gave apparent $K_D$ values that were up to seven-fold higher than the actual $K_D$ values. A more extreme literature example is discussed in the next section, with discrepancies reaching 1000-fold, and other examples have been previously noted (*Hulme and Trevethick, 2010*; *Strohkendl et al., 2018*). There is a tendency to be less careful about controls in pursuit of relative affinities (specificity) rather than absolute affinity. However, failing to account for the factors noted above can also underestimate specificity by orders of magnitude (see *Figure 4—figure supplement 1* and *Figure 5—figure supplement 4* below).

These observations highlight an urgent need to revisit the criteria for reliable binding measurements. There is a parallel need to render these criteria accessible to a broad range of biologists, regardless of background or training, in the form of clear and readily actionable guidelines. To meet these needs, we provide simple, concrete strategies so that any practitioner can carry out reliable binding measurements, clearly communicate their results, and evaluate results from others.

Fortunately, the key requirements for binding measurements can be broken down into a small number of steps. We present two required steps for equilibrium binding measurements—varying the incubation time (see section 'Vary incubation time to test for equilibration') and controlling for titration (see section 'Avoid the titration regime'), and we illustrate these steps for the example of RNA binding to the *Saccharomyces cerevisiae* Puf4 protein (*Gerber et al., 2004*; *Miller et al., 2008*). We also present additional steps that can be taken to further increase confidence in $K_D$ values and to obtain kinetic information about the binding event under investigation (see sections 'Test $K_D$ by an independent approach' and 'Determine the fraction of active protein'). Finally, we describe strategies to address cases where no binding is initially detected and explain why it is often premature to conclude an absence of binding (see section 'The case of no observed binding').

## Practical considerations

In principle, one would like to have well-behaved and perfectly controlled measurements in all cases, but biology and biochemistry can be messy. There are many times, working with extracts and partially purified systems where protein concentrations cannot be accurately determined, where proteases and nucleases may limit achievable equilibration times, and where there may be additional interacting components. Regardless of these potential complications, the simple steps indicated

below can establish the robustness of measured affinities and can diagnose and help overcome issues like loss of activity over time. Moreover, these controls (and quantitative measurements more generally) can help uncover new features and regulatory mechanisms, based on deviations from 'ideal' behavior of simple binding equilibria.

### Vary incubation time to test for equilibration

The most basic test for whether a binding reaction has reached equilibrium is that the fraction of complex formed between two molecules does not change over time. Nevertheless, the majority of papers we surveyed that present binding measurements and report apparent affinities or equilibrium dissociation constants do not report that time has been varied (*Figure 1*). We first describe two related concepts that will help readers develop an intuition for the time scales of binding processes and we then apply these concepts to Puf4 binding.

#### Half-life

Binding and other simple kinetic processes, in general, follow exponential curves (*Figure 2*). The key property of an exponential curve is that it has a constant half-life ($t_{1/2}$)—that is, the time it takes for the reaction to proceed from 0% to 50% complete, 50% to 75% complete, 75% to 87.5% complete, etc. is the same (*Figure 2*). After three half-lives, an exponential process is almost 90% complete ($3t_{1/2} = 87.5\%$; *Figure 2*), which is close enough to equilibration for most applications. Below we adopt the more common standard of taking reactions to five half-lives, or 96.6% completion; this more conservative standard is safer given that there are multiple sources of potential error in practice.

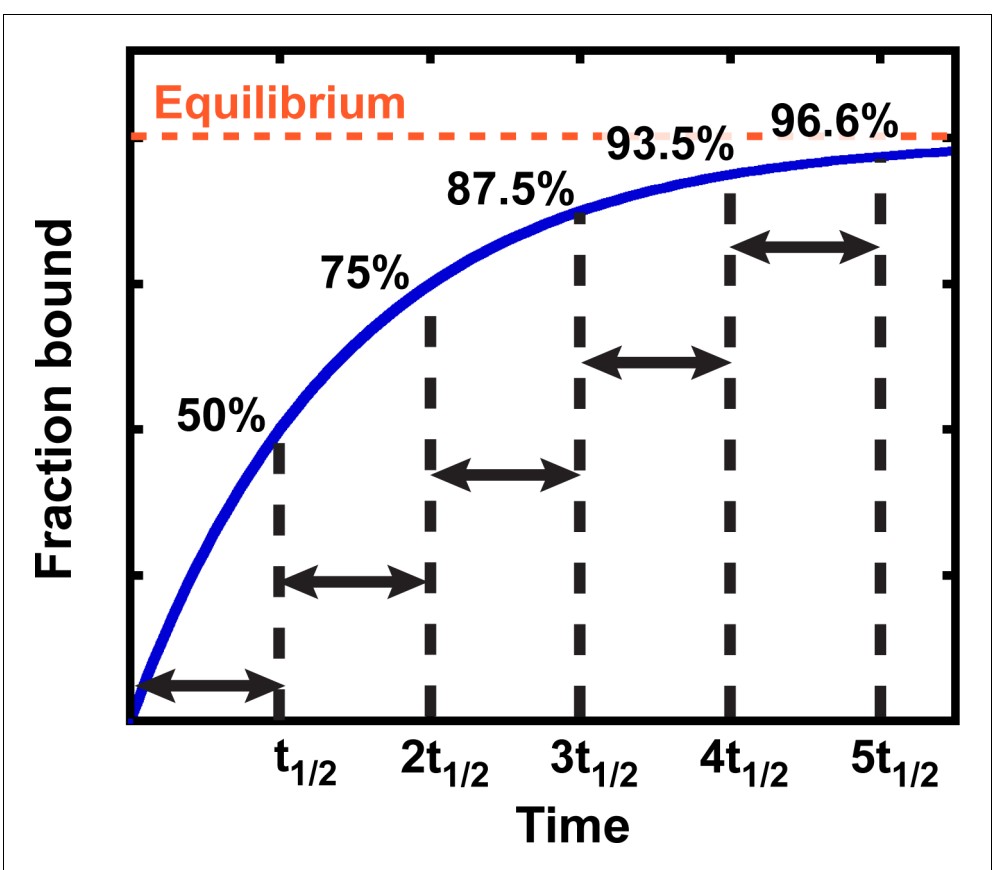

**Figure 2.** Exponential kinetics used to estimate the time needed for binding equilibration. Arrows indicate reaction half-life $t_{1/2}$. Fraction bound is defined by the equation $1-e^{-t\times\ln2/t_{1/2}} = 1-e^{-t\times k_{equil}}$.

## Equilibration rate constant

The equilibration rate constant is effectively the inverse of the binding half-life ($k_{\text{equil}} = \frac{\ln 2}{t_{1/2}} \approx \frac{1}{t_{1/2}}$) and, importantly, is concentration-dependent. For the binding equilibrium shown in *Figure 3*, under conditions where one binding partner (here, the protein, P) is in large excess over the other (RNA), the rate equation for approach to equilibrium, $k_{\text{equil}}$, is described as:

$$k_{\text{equil}} = k_{\text{on}}[\text{P}] + k_{\text{off}} \tag{1}$$

$k_{\text{on}}$ is the association rate constant, [P] is the concentration of protein, or the binding partner in excess, and $k_{\text{off}}$ is the dissociation rate constant (*Pollard, 2010*). According to *Equation 1*, equilibration is the slowest at the lowest protein concentrations. For this reason, equilibration times need to be established from the low end of the concentration range. In practice, it is useful to consider the limiting case with the protein concentration approaching zero ([P] ~ 0), such that *Equation 1* simplifies to *Equation 2* (*Hulme and Trevethick, 2010*):

$$k_{\text{equil,limit}} = k_{\text{off}} \tag{2}$$

Thus, the more long-lived the complex (i.e. the lower its dissociation rate constant), the longer the incubation time required to reach equilibrium.

What is the range of equilibration times for typical biomolecular interactions? While $k_{\text{off}}$ measurements (and, consequently, $k_{\text{equil}}$) are less common in literature than $K_D$ measurements, equilibration times can be readily estimated (*Sanders, 2010*). Given that $K_D = \frac{k_{\text{off}}}{k_{\text{on}}}$ (*Figure 3*) and assuming that the binding of molecules occurs as fast as diffusional collisions ($k_{\text{on}} = 10^8$ M$^{-1}$s$^{-1}$), we can calculate that an interaction with a $K_D$ value of 1 pM would require a 10 hr incubation to reach equilibrium, whereas a 1 μM $K_D$ interaction would only require 40 ms (*Table 1*). Notably, binding rate constants for processes involving macromolecules are often smaller than the diffusion driven limit of ~$10^8$ M$^{-1}$s$^{-1}$, for example when additional conformational rearrangements are required for stabilizing binding after two molecules collide (*Karbstein and Herschlag, 2003*; *Peluso et al., 2000*; *Wu et al., 2002*). As a result, equilibration can take much longer. Thus, equilibration times for two interactions with the same $K_D$ value can vary by orders magnitude, and some reactions in the biologically relevant affinity range can require equilibration times of 10s of hr or even longer in vitro (*Table 1*; *Hulme and Trevethick, 2010*; *Sanders, 2010*). These long times underscore that biology has developed mechanisms to circumvent or utilize such slow processes—for example, rapid association may

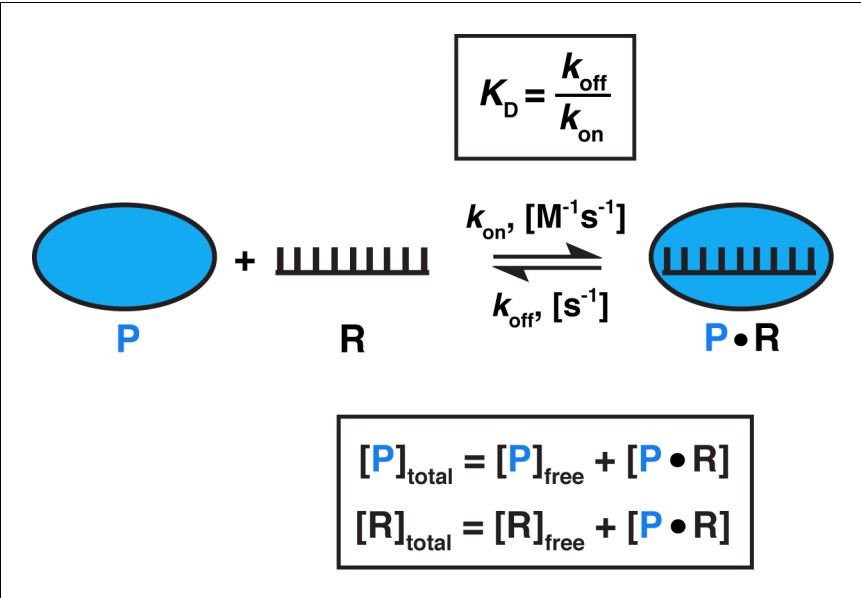

**Figure 3.** Model for one-step, non-cooperative, 1:1 binding between two molecules. Protein (P) binding to an RNA (R) molecule is shown for illustrative purposes.

**Table 1.** Equilibration times ($t_{equil}$) for different affinities and association rate constants.

| $K_D$ | $k_{on}$, $M^{-1}$ $s^{-1}$ | $t_{equil}$* | |
|---|---|---|---|
| | | sec | hr |
| 1 µM | $10^8$ | 0.04 | |
| | $10^6$ | 4 | |
| | $10^3$ | | 1 |
| 1 nM | $10^8$ | 40 | |
| | $10^6$ | | 1 |
| | $10^3$ | | 1000 |
| 1 pM | $10^8$ | | 10 |
| | $10^6$ | | 1000 |
| | $10^3$ | | 1,000,000 |

*$t_{equil}$ was calculated as five half-lives: $t_{equil} = 5t_{1/2} = 5 \times 0.693/k_{equil}$, where $k_{equil} = k_{off} = K_D \times k_{on}$ (**Equation 2** and **Figure 3**).

be facilitated by high intracellular concentrations of binding partners, and cellular factors such as molecular chaperones, helicases, chromatin remodelers, or translation can speed up binding and dissociation.

## Implications of insufficient equilibration

Despite the realistic possibility of long equilibration times for biological association events, nearly 90% of the reported incubation times were 1 hr or less (*Figure 1—figure supplement 1B*). As a concrete example, several 'equilibrium' dissociation constants reported for CRISPR nucleases, which are well known for tight RNA and/or DNA binding, were determined from incubations of 1 hr or less (e.g. *Semenova et al., 2011*; *Westra et al., 2012*; *Westra et al., 2013*; *Sternberg et al., 2014*; *O'Connell et al., 2014*; *Wright et al., 2015*; *Ma et al., 2015*; *Jiang et al., 2015*; *Sternberg et al., 2015*; *Beloglazova et al., 2015*; *Rutkauskas et al., 2015*; *Abudayyeh et al., 2016*; *Supplementary file 2*). But when target dissociation of these proteins was measured over time, it took many hours (*Strohkendl et al., 2018*; *Richardson et al., 2016*; *Boyle et al., 2017*; *Raper et al., 2018*), suggesting that equilibration takes much longer than an hour and that the reported $K_D$ values based on these short incubation times underestimate the true binding strength. In one striking example, kinetic measurements revealed an equilibration time of >100 hr for the Cas12a complex and an equilibrium constant that was 1000-fold lower than previously reported for the same enzyme at similar conditions after much shorter incubation time (*Strohkendl et al., 2018*). Insufficient incubation times for tight binders may have also led to underestimation of specificity, a topic of central concern for CRISPR targeting (and for much of biology). *Figure 4—figure supplement 1* illustrates how target affinities that differ by two orders of magnitude may appear identical if the incubation time is too short.

An example in which extending the incubation time changed the mechanistic interpretation comes from studies of the signal recognition particle (SRP). Originally, the observation that 4.5S RNA enhanced the assembly of the signal recognition particle (SRP) and SRP receptor led to a proposed mechanism in which the 4.5S RNA stabilized the complex. Subsequently, binding studies extended to longer times revealed that the 4.5S RNA accelerated the otherwise slow SRP/receptor binding and dissociation *without* affecting the binding affinity (*Peluso et al., 2000*). Exploring the time dependence of the assembly process changed the mechanistic conclusions: 4.5S RNA could be shown to play a catalytic, rather than stabilizing role in SRP/receptor assembly.

*Figure 4—figure supplement 1* illustrates how incubation times that are very far from equilibrium can lead to systematic deviations of the data from the fit to an equilibrium binding equation. While a poor fit is not sufficient to diagnose insufficient equilibration (and, conversely, a good fit does not prove complete equilibration), an inability to fit the data well to a simple binding model provides an important indicator that additional controls are required. Only after simple controls for equilibration and titration (see below) have been performed, should more complex binding models, such as

cooperativity, be considered, unless such models are independently supported. Indeed, among the studies in our literature survey omitting one or both key controls, several included poorly fit binding curves. Importantly, graphs of fits of the data to a clearly defined equilibrium binding model should be published along with the $K_D$ values when possible, and the quality of the fit over the entire concentration range should always be carefully assessed. In summary, the incubation time must be varied to ensure equilibration, ideally across a range of at least 10-fold. Below we illustrate this control, and the need for it, with experimental results for Puf4 binding to its consensus RNA.

## Time dependence of Puf4 binding at 25°C and 0°C

To establish the equilibration time for Puf4 binding to its cognate RNA sequence, Puf4 was mixed, over a series of concentrations, with a trace amount of labeled RNA (in this case, $^{32}$P-labeled; 0.002–0.016 nM) and incubated for a specified time ($t_1$) (*Figure 4A*). The fraction of bound RNA was subsequently determined by non-denaturing gel electrophoresis (see Materials and methods).

At 25°C, we observed the same amount of binding with incubations of $t_1$ = 30 min, 1.5 hr, and 4.5 hr at each protein concentration, providing strong evidence for equilibration even at the shortest time (*Figure 4B*). Consequently, we can proceed to the next key control at this condition, using an incubation time of ≥30 min.

We also present Puf4 binding results at 0°C as these data provide an example of slow equilibration and because many binding studies report incubations on ice to stabilize binding. Indeed, the results at 0°C were very different than those at 25°C. As shown in *Figure 4C*, Puf4 bound different amounts of RNA in the 30 min, 1.5 hr, and longer incubations. Not until the incubation was extended to 4.5 hr did the extent of binding level off at the lowest Puf4 concentrations—that is, the amount bound was the same after 4.5 and 24 hr. Consequently, equilibration of Puf4–RNA binding on ice requires *at least* 4.5 hr, and incubation for only 30 min would give an apparent $K_D$ value that is seven times higher than after a 24 hr incubation. Moreover, binding at 0°C was so tight that we were only able to obtain part of the binding curve while maintaining the protein concentrations in excess of labeled RNA (*Figure 4C*). The importance of this excess to obtain reliable $K_D$ values is described in the next section. In the 0°C case and more generally, it is important to re-assess the equilibration time after establishing that binding is in an appropriate concentration regime, as we demonstrate in later sections. Similarly, changes in conditions, such as salt concentration, temperature or pH, can affect both the affinity and the equilibration time and therefore should be accompanied by confirming that equilibration has occurred.

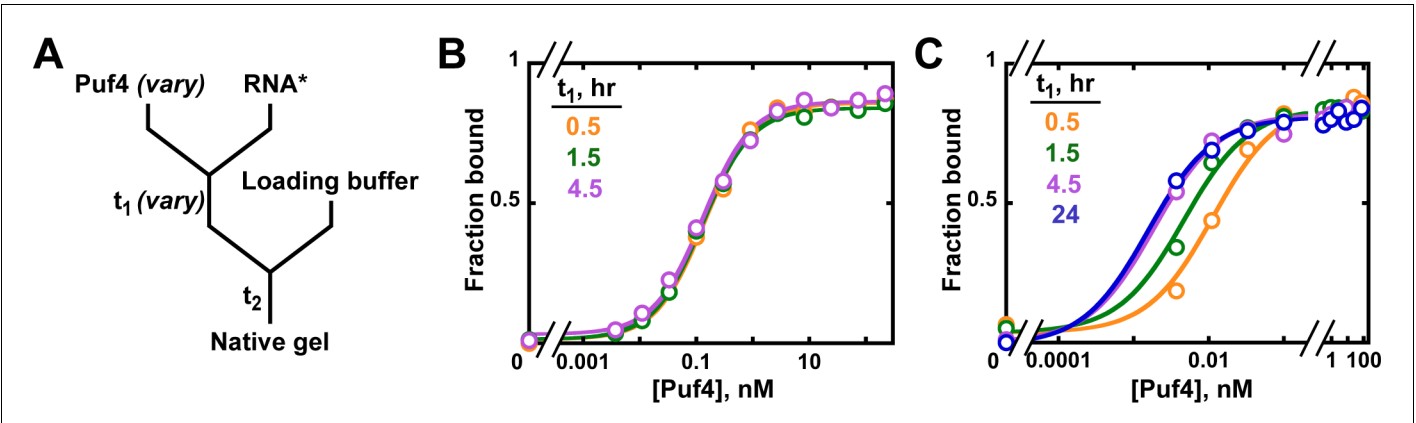

**Figure 4.** Establishing equilibration in affinity measurements. (A) Mixing scheme. RNA*: labeled RNA (here—5′-terminally labeled with $^{32}$P). In addition to varying equilibration time $t_1$ (main text), the time and conditions between adding the loading buffer and loading ($t_2$) are controlled (see Appendix 2—note 2). (B, C) Concentration dependence of Puf4 binding at 25°C (B) and at 0°C (C) after different incubation times. Data were collected at protein concentrations greater than or equal to the concentration of labeled RNA (0.002–0.016 nM, indicating the lower and upper limit of labeled RNA concentration; see section 'Avoid the titration regime' and Appendix 2—note 4).

The online version of this article includes the following figure supplement(s) for figure 4:

**Figure supplement 1.** Insufficient equilibration times can lead to incorrect determination of relative affinities.

## Avoid the titration regime

The most common approach to measuring affinity is to vary the concentration of one component, while keeping the concentration of the other binding partner constant. However, this experimental design is not always sufficient, as there are two limiting regimes, determined by the concentration of the constant component; only one of these concentration regimes allows the $K_D$ to be reliably determined, while the other does not.

In the first, 'binding' regime, the concentration of the constant ('trace') component, R, is well below the dissociation constant ([R]$_{total}$ << $K_D$ for the example in *Figure 3*). In this case, the concentration of the variable component (P in *Figure 3*) that gives half binding is equal to the $K_D$

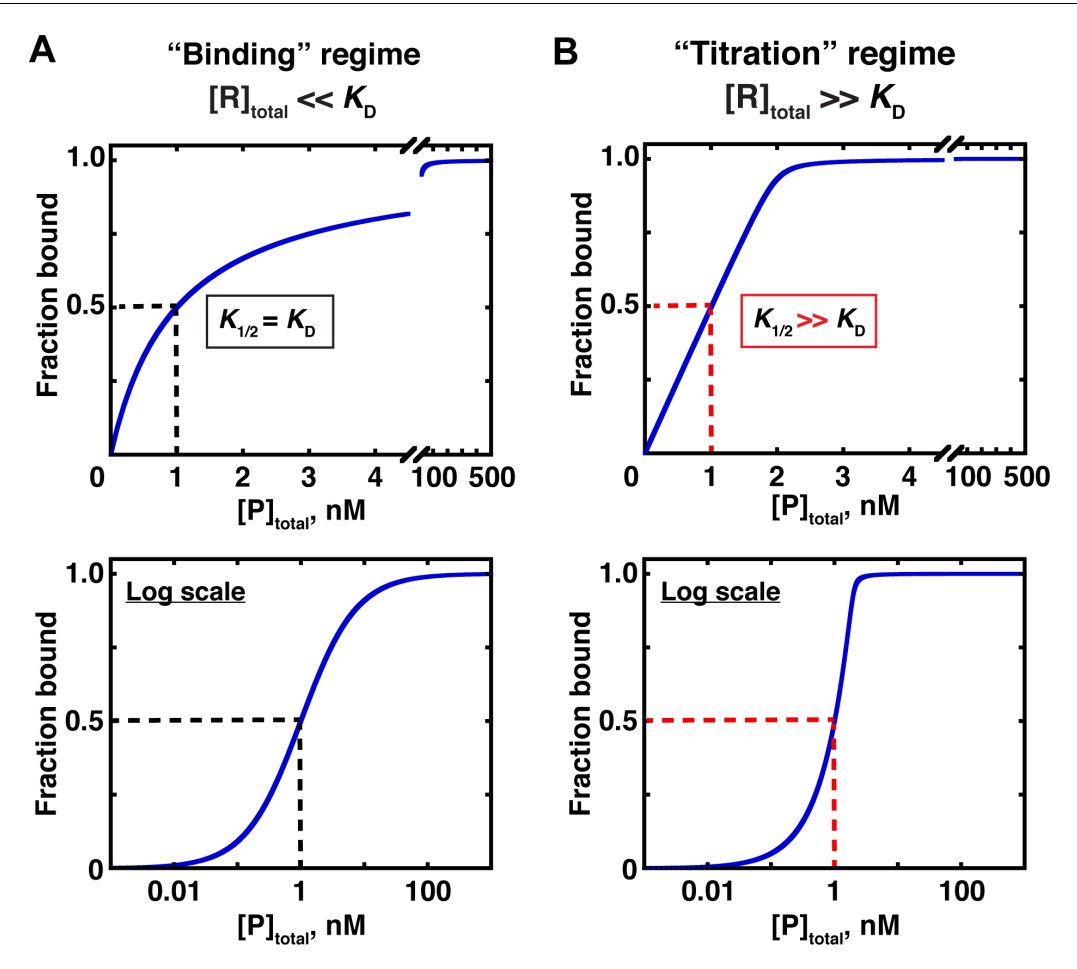

**Figure 5.** Two concentration regimes. (A) Binding curve for the model in *Figure 3* in the 'binding' regime—that is, the trace binding partner concentration ([R]$_{total}$) is much lower than $K_D$ and much lower than [P]$_{total}$ (*Equation 4b*). Here, the $K_D$ is simply the protein concentration at which half of the RNA is bound ($K_{1/2}$, here corresponding to 1 nM). The same simulated binding curve is shown in linear (top) and log (bottom) plots, as both are useful and common in the literature. (B) Binding curve in the 'titration' regime, simulated for an interaction with a $K_D$ value of 0.01 nM and an [R]$_{total}$ of 2 nM. Although the $K_{1/2}$ value in this example is identical to the example in Part A, here it does not equal $K_D$, instead exceeding the real $K_D$ value by 100-fold.

The online version of this article includes the following figure supplement(s) for figure 5:

**Figure supplement 1.** The effects of RNA (ligand) concentration on observed binding.

**Figure supplement 2.** Fit to the quadratic binding equation becomes less sensitive to differences in $K_D$ when the RNA concentration is in large excess over the $K_D$.

**Figure supplement 3.** Application of the hyperbolic (*Equation 4b*) and quadratic (*Equation 5*) binding equations to simulated binding data with increasing noise levels.

**Figure supplement 4.** Effects of trace binding partner concentration on apparent relative affinities.

**Figure supplement 5.** Concentration regimes that do not (A) and do (B) affect the determination of equilibrium binding constants.

(*Figure 5A*). In the other, 'titration' regime, the concentration of the constant component is much greater than the $K_D$ ($[R]_{total} \gg K_D$) so that essentially all added P is depleted from solution due to binding to R, until there is no more free R left to bind. In this case, the concentration of P that gives half binding does not equal or even approximate the $K_D$. Rather, at high excess of R over the $K_D$, the concentration of P that gives half binding is simply half of the concentration of (active) R molecules—a value that can differ from the sought-after $K_D$ by orders of magnitude (*Figure 5B*; *Figure 5—figure supplement 1*).

A potentially useful intermediate regime exists between the two extremes, with limiting component concentrations similar to or in modest excess over the $K_D$. The $K_D$ can be determined in this regime by using an appropriate binding equation, although with potential pitfalls (see below).

## Distinguishing between concentration regimes

The challenge is that distinguishing between the regimes requires the knowledge of the $K_D$, and consequently it is impossible to know a priori which regime holds. A useful rule of thumb for avoiding the titration regime is to always maintain the concentration of the excess binding partner significantly above that of the trace limiting partner. The reason for this can be gleaned from the equation that describes the fraction of bound RNA for the simple binding scheme of *Figure 3*:

$$\text{Fraction bound} = \frac{[P]_{free}}{[P]_{free} + K_D} \tag{4a}$$

Here $[P]_{free}$ is the unbound protein concentration and $K_D$ is simply the free protein concentration at which half of the RNA is bound. But while *Equation 4a* holds universally, in practice we only know the *total* concentration of P, $[P]_{total}$—how much we added to the solution—not the free concentration ($[P]_{free}$). Therefore, we want to operate under simplifying conditions where $[P]_{free} \approx [P]_{total}$ so that we can substitute $[P]_{total}$ into *Equation 4a* to give *Equation 4b*:

$$\text{Fraction bound} = \frac{[P]_{total}}{[P]_{total} + K_D} \tag{4b}$$

The condition $[P]_{free} \approx [P]_{total}$ holds true if P is in large excess of RNA across the entire experiment, meaning that only a small fraction of total protein is used up by binding to RNA. Most importantly, this condition must hold for the protein concentration that gives half-saturation to determine the $K_D$; hence the requirement for the binding regime that the concentration of the limiting component must be $\ll K_D$. Nevertheless, simply maintaining an excess of protein over the limiting component may not always be sufficient to maintain a binding regime, given the uncertainty often surrounding concentration measurements and even greater uncertainty surrounding active concentrations.

In principle, a more complex quadratic binding equation provides an alternative to working under the $[P]_{free} \approx [P]_{total}$ assumption, as it explicitly accounts for bound protein:

$$\text{Fraction bound} = \frac{([R]_{total} + [P]_{total} + K_D) - \sqrt{([R]_{total} + [P]_{total} + K_D)^2 - 4 \times [R]_{total} \times [P]_{total}}}{2 \times [R]_{total}} \tag{5}$$

Indeed, several techniques (most notably ITC) commonly operate outside the binding regime and rely on *Equation 5* (or equivalent formulations) for data fitting. Importantly, the quadratic equation is only applicable to the intermediate and binding regimes, but not the titration regime. The reason for this is that at very high concentrations relative to the $K_D$, the contribution of $K_D$ in determining the fraction bound (*Equation 5*) becomes negligible, and as a result a meaningful $K_D$ value cannot be extracted from the fit to the binding data. Simulated data in *Figure 5—figure supplements 2* and *3* illustrate this limitation. Consequently, even when using *Equation 5*, the concentration of the limiting component should be kept to a minimum to avoid the titration regime.

Where does the intermediate regime end and titration begin? The answer depends on the technique and the quality of the data. For ITC measurements, which provide highly precise information for each added binding aliquot, up to 1000-fold excess of the limiting species over the measured $K_D$ can be acceptable (*Velázquez-Campoy et al., 2004*). However, in most other cases, this limit is

much lower. Simulations in *Figure 5—figure supplement 3* suggest that up to ~10-fold excess consistently allows for reasonably well-defined $K_D$ values in the presence of typical binding data, and up to 100-fold excess can be useful for data with minimal noise. In contrast, performing the experiments in the binding regime (fit with *Equation 4b*) yields well-defined $K_D$ values even with substantial noise in the data (*Figure 5—figure supplement 3*).

## Implications of the titration regime

Of the 100 literature studies we surveyed, most (65%) determined $K_D$ values under the assumption of the binding regime, by using *Equation 4b* or equivalent analysis. Nevertheless, the required condition that the limiting species concentration be $<<K_D$ was not always supported. One-third of the studies using *Equation 4b* (n = 21) reported $K_D$ values that were comparable to (<10-fold excess) the concentration of the trace component, including nine studies in which the reported $K_D$ was indistinguishable from (within ~2-fold) or even below the stated trace component concentration, consistent with an intermediate or even titration regime (*Figure 1—figure supplement 2*).

The implication in all these cases is that the reported $K_D$ values may underestimate the real affinities. Unfortunately, it is difficult to determine the extent of this underestimation post-factum without further experimental controls. To understand why, recall from the example in *Figure 5B* that in the titration regime the midpoint of the binding curve only reflects ~half the concentration of the limiting species, which sets a lower limit to the apparent $K_D$ derived from *Equation 4b*, even if the real $K_D$ is much lower. Conversely, if the midpoint of the binding curve (and the reported $K_D$ in the above cases) is approximately the same as the limiting component concentration (allowing for some uncertainty in the concentration), the real $K_D$ could be anything below this value, from several-fold to many orders of magnitude less. As with insufficient incubation, systematic deviations of the data from the fit to *Equation 4b* can be a clear indicator that the apparent $K_D$ is limited by titration, but a good fit should not be considered sufficient to prove the binding regime, as experimental uncertainties and other causes can mask deviations.

High-affinity interactions are most susceptible to titration, a corollary of the simple fact that for very low $K_D$ values it becomes increasingly difficult to maintain concentrations much lower than $K_D$ while still allowing for detection. Since CRISPR nucleases represent some of the most widely studied high-affinity binders, we surveyed a sample of studies to determine the concentration regime under which the reported $K_D$ values were measured (*Supplementary file 2*). Of the 15 studies, the majority (13, or 90%) assumed the binding regime in their analysis, indicated by the use of *Equation 4b* or equivalent. However, only two of these studies (15%) reported using labeled DNA or RNA concentrations considerably below the apparent $K_D$, and in five cases the lowest reported $K_D$ was essentially identical to the labeled RNA or DNA concentration (within ~2-fold), consistent with possible titration.

Importantly, because relative affinities are typically based on the tightest binders, titration effects on the 'wild-type' substrate measurements can distort all specificity (relative affinity) values that are based on it. *Figure 5—figure supplement 4* illustrates an example, in which two substrates with a 100-fold difference in affinity appear to have identical or near-identical affinities when titration is not controlled for.

Given the impossibility of designing experiments for the binding regime a priori, without knowing the affinity, it is important to rule out titration empirically. Thus, analogously to varying time to establish equilibrium, we strongly recommend systematically varying the concentration of the limiting species to establish the binding regime (or, with use of *Equation 5*, the intermediate regime). The hallmark of a valid $K_D$ is that it is not affected by varying the concentration of the limiting component, whereas a titration regime would result in concentration-dependent apparent $K_D$ values. At a minimum, this control should always be performed when the measured $K_D$ value is comparable to the concentration of the limiting component (*Equation 4b*), or when *Equation 5* yields poorly defined apparent $K_D$ values or values much lower than the limiting concentration. Below we demonstrate the titration control for Puf4 affinity measurements.

## RNA concentration dependence of Puf4 binding at 25°C and 0°C

We systematically varied the labeled RNA concentration in Puf4 binding experiments at 25°C and 0°C, to illustrate the binding and intermediate regimes, respectively. *Figure 5—figure supplement 5* provides a schematic description of the two regimes to help build the reader's intuition.

At 25°C, the Puf4 binding curves were identical across a nine-fold range of RNA concentrations (*Figure 6A,B*), and the data were well described by *Equation 4b*. From the constancy of the binding curves in *Figure 6B*, we can conclude that the binding regime holds for Puf4 at 25°C, and thus that the observed $K_D$ value of 120 pM obtained from *Equation 4b* represents a true equilibrium constant. As expected for the binding regime, the measured $K_D$ is higher than the RNA concentrations (120 pM vs 2–18 pM).

The situation is different at 0°C (*Figure 6C*). Here, varying the labeled RNA concentration revealed divergent binding curves and a pronounced dependence of apparent affinity (determined by fitting the data to *Equation 4b*) on the concentration of RNA, the constant component (*Figure 6C,D*). Moreover, the fits of the data to *Equation 4b* (solid lines in *Figure 6C*), which assumes $[P]_{free} \approx [P]_{total}$, were poor, increasingly so for higher RNA concentrations. These data are indicative of protein depletion due to binding to labeled RNA. The apparent $K_D$ values vary by five-

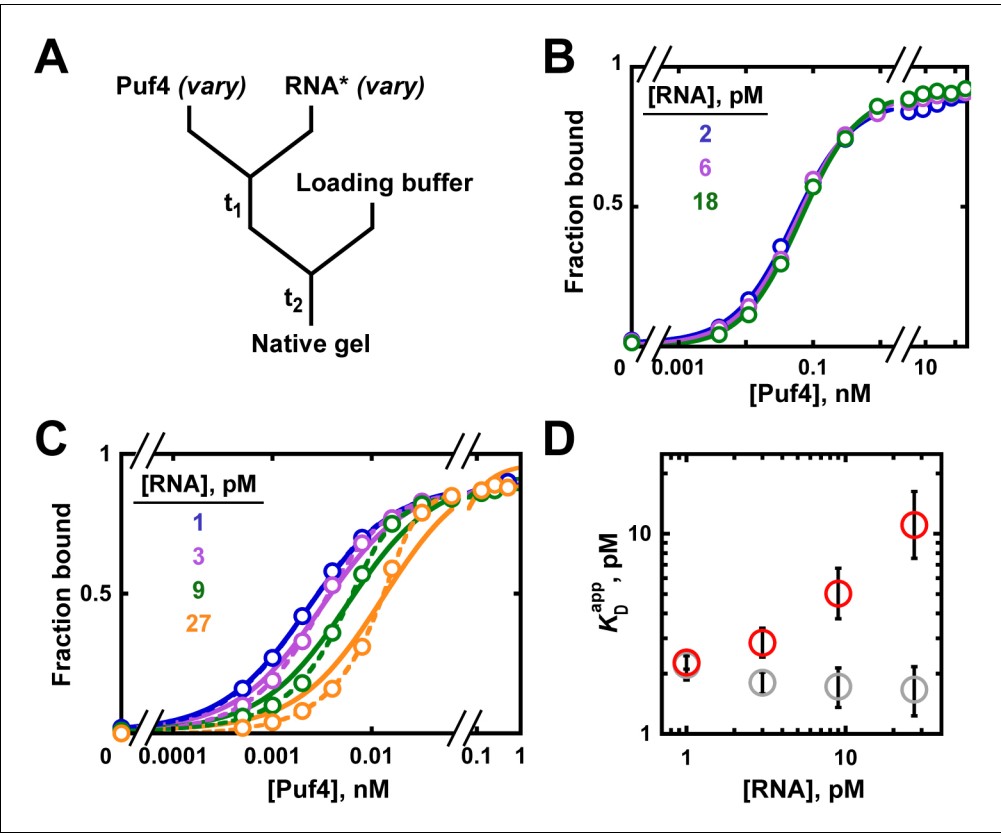

**Figure 6.** Varying the concentration of the 'trace' binding partner. (A) Mixing scheme, as in *Figure 4A* but now with a series of labeled RNA concentrations. (B) Puf4 binding to different concentrations of [32]P-labeled RNA at 25°C. For simplicity, only the lower limits of RNA concentration are indicated; the corresponding upper limits were 15–140 pM RNA (see Materials and methods and Appendix 2—note 4). Incubation time $t_1$ was 0.5 hr, as established in *Figure 4B*. (C) Puf4 binding to different concentrations of [32]P-labeled RNA at 0°C. Lower limits of labeled RNA concentration are indicated. Incubation time $t_1$ was 40 hr. Note that these data are not fit well by *Equation 4b*, which assumes $[R^*]_{total} \ll K_D$ (solid lines). Quadratic fits, which do not assume negligible RNA concentration, are shown in dashed lines (*Equation 5*). (D) Effect of RNA concentration on apparent $K_D$ ($K_D^{app}$) at 0°C. Red symbols indicate $K_D^{app}$ values from a hyperbolic fit (*Equation 4b* and solid lines in *C*) and grey symbols indicate $K_D^{app}$ values from fits to the quadratic equation (*Equation 5*). The error bars denote 95% confidence intervals, as determined by fitting the data to the indicated equation in Prism 8.

fold across the 30-fold range of RNA concentrations used (*Figure 6D*, red circles), and even greater discrepancies would arise at higher RNA concentrations (*Figure 5—figure supplement 1*). Consequently, only an upper limit of the real affinity can be extracted from these data ($K_D \leq 2.3$ pM, based on the fit value at the lowest RNA concentration used).

To address the limitation in our 0°C data we could, in principle, lower the concentration of labeled RNA even further, until the labeled RNA concentration is $<<K_D$ and until an RNA concentration-independent regime is established. But this is difficult when binding is very tight, as a limit is set by the sensitivity of the technique used. In our case, at ~1 pM $^{32}$P-labeled RNA we are already near the limit of reliable detection. If the concentration of the trace component cannot be lowered further, a more sensitive approach can sometimes be found. Kinetic approaches are particularly suitable for tight binders (see Appendix 1), or one can report an upper limit of the $K_D$. In some cases, increasing the salt concentration or other changes to the solution or binding partners can be used to weaken binding to make it easier to obtain affinities at higher concentrations of the labeled species; this approach can be especially valuable if one is primarily interested in the relative affinities of multiple ligands (*Altschuler et al., 2013*).

As noted earlier, the quadratic binding equation enables $K_D$ determination for binding reactions in the intermediate regime. The quadratic equation provides a good fit to the 0°C data (*Figure 6C*, dashed lines) and yields uniform and well-defined $K_D$ values of ~1.9 pM across the different RNA concentrations, consistent with an intermediate (rather than titration) regime (*Figure 6D*, grey circles). The same $K_D$ value was obtained from kinetic experiments, providing independent support for and confidence in this determination (Appendix 1).

In summary, we want to use the binding regime whenever possible, as it allows for the most straightforward and reliable $K_D$ measurements. It is necessary to avoid the titration regime and caution is required in the intermediate regime. In practice, varying the concentration of both components is an essential control for ruling out titration, ruling out other potential artifacts, and ensuring the measurement of valid dissociation constants.

## Re-evaluating the equilibration time at 0°C

In the previous section, we mentioned the need for re-evaluating the equilibration time for Puf4 binding at 0°C after a binding regime was established. In principle, after determining sufficiently low RNA concentration for the binding regime, one could vary the incubation time again, as done in *Figure 4*. In our case, we used the shortcut defined in *Equation 2* and instead determined the upper limit of the equilibration time by measuring the $k_{off}$ at 0°C (Appendix 1; see also Appendix 2—note 1 for precautions when applying this shortcut). These measurements revealed an equilibration time of 30 hr (five half-lives), far above the typical incubation times of 1 hr or less (*Figure 1—figure supplement 1*).

## Dependence of binding affinity on conditions

The 100-fold difference in Puf4 affinity between 0°C and 25°C underscores the important point that the equilibrium dissociation constant is only a constant value at a given set of conditions, and that the affinity can change dramatically when the conditions (temperature, salt, pH) are changed. This dependence on conditions should always be considered when comparing literature values or when applying in vitro results to biology.

## Test $K_D$ by an independent approach

Even when no challenges are encountered, as in the case of Puf4 binding at 25°C, it is a good idea to determine the $K_D$ by a second approach to ensure that the measurement is not biased by experimental artifacts or idiosyncrasies of a particular technique. This is especially important when using a secondary readout (vs. a direct approach) such as native gel shift or nitrocellulose filter binding, where major loss (or gain) of bound complex can potentially occur between the equilibration and detection steps (see below and Appendix 2—note 2).

Of course, there are many approaches to carrying out equilibrium binding measurements one can choose from (e.g. *Velázquez-Campoy et al., 2004*; *Wong and Lohman, 1993*; *Eftink, 1997*; *McDonnell, 2001*). Here, we used a kinetic approach for independent $K_D$ determination for Puf4 at 25°C and 0°C, as described in Appendix 1. Kinetic measurements provide an information-rich

alternative and complement to the equilibrium measurements and are often simple to carry out provided they fall within a measurable time range (*Pollard, 2010*; *Hulme and Trevethick, 2010*; *Sanders, 2010*; *Pollard and De La Cruz, 2013*). In case of Puf4, the affinities determined by kinetic measurements were within two-fold of those from equilibrium determinations, strongly supporting their accuracy.

## Determine the fraction of active protein

The amount of bound ligand is determined not by the total protein concentration but by the concentration of total *active* protein. If 90% of the protein is damaged due to misfolding, aggregation, degradation or, for example, inactivated by phosphorylation at the binding interface, then the observed affinity will be that for only 10% of the total protein present—and will be ten-fold higher than the actual $K_D$ value. Moreover, if the binding-competent protein concentration is much lower than the total and therefore much closer to the limiting component concentration than expected, the binding regime may not be maintained, leading to even greater discrepancies between the real and observed $K_D$. As a common cause of non-active or less active protein is aggregation, determining the monodispersity of the protein following purification is advisable (*Altschuler et al., 2013*).

In addition, we recommend, when possible, a titration experiment to determine the fraction of binding-competent protein (*Altschuler et al., 2013*). Here, a concentration of ligand that is much greater than the measured $K_D$ is intentionally used and the protein concentration is varied by approximately an order of magnitude above and below the ligand concentration. To ensure accurate ligand concentration and to prevent excessive signal (if labeled ligand is used), the trace labeled ligand should be mixed with a large excess of identical unlabeled molecule at a known concentration. Assuming that the stoichiometry of the bound complex is known and that the ligand is 100% active, the breakpoint in fraction bound versus the ratio of protein to ligand indicates the amount of active protein (*Figure 7*). For example, for a 1:1 complex, a breakpoint at a protein:RNA ratio of 2.0 suggests that half of the protein is active. In *Figure 7*, the ratio of 1.3 suggests that the Puf4 preparation is 75% active (0.75 = 1/1.3). Consequently, the apparent $K_D$ values determined in the previous sections should be multiplied by the active protein fraction (which ranged from 0.75 to 0.90 for Puf4) to determine the final $K_D$ value. In an alternative approach, the titration data could be fit to a quadratic equation, with a coefficient used to represent the active protein fraction (*Figure 7—figure supplement 1*).

A limitation of the titration experiment is that it assumes the constant component to be 100% active, which may not always be the case, especially in the case of protein-protein interactions. Therefore, one should ensure, to the extent possible, maximum purity of both binding components. Importantly, one should always make clear whether experiments were carried out to determine 'fraction active'.

## The case of no observed binding

Researchers often conclude that there is 'no binding'—that 'X does not bind to Y'. Typically, the underlying experimental observation is an absence of observed binding up to a certain protein (or ligand) concentration. Therefore, one should report a lower limit for the dissociation constant ($K_D$), rather than draw an absolute conclusion of 'no binding'. But even an accurate lower limit often requires additional experiments, because the absence of observed binding—say in a gel shift, filter binding, or pull-down experiment—can arise either because there is no significant binding or because the complex does not withstand the assay conditions (*Pollard, 2010*). While this objection may seem like a technicality, there are many instances where known binders do not give a gel shift or filter binding.

Immuno-precipitation and pull-down assays are pervasive in current biological investigations and are often interpreted in terms of 'binding' or 'no binding'. But the reality of the interpretation of these experiments—and the reality of molecular interactions—is more nuanced (*Pollard, 2010*). A ligand with the same affinity, slightly lower affinity, or even higher affinity than another ligand with demonstrated binding can incorrectly be concluded to 'not bind'.

Consider, for example, an RNA pull-down with an RNA binding protein with $K_D = 10^{-9}$ M and $k_{on} = 10^8$ M$^{-1}$ s$^{-1}$; this gives $k_{off} = 0.1$ s$^{-1}$ or a half-life for dissociation of ~10 s. If the washing steps following a pull-down take 30 s, only ~10% of the complex is expected to remain. If the affinity is 10-

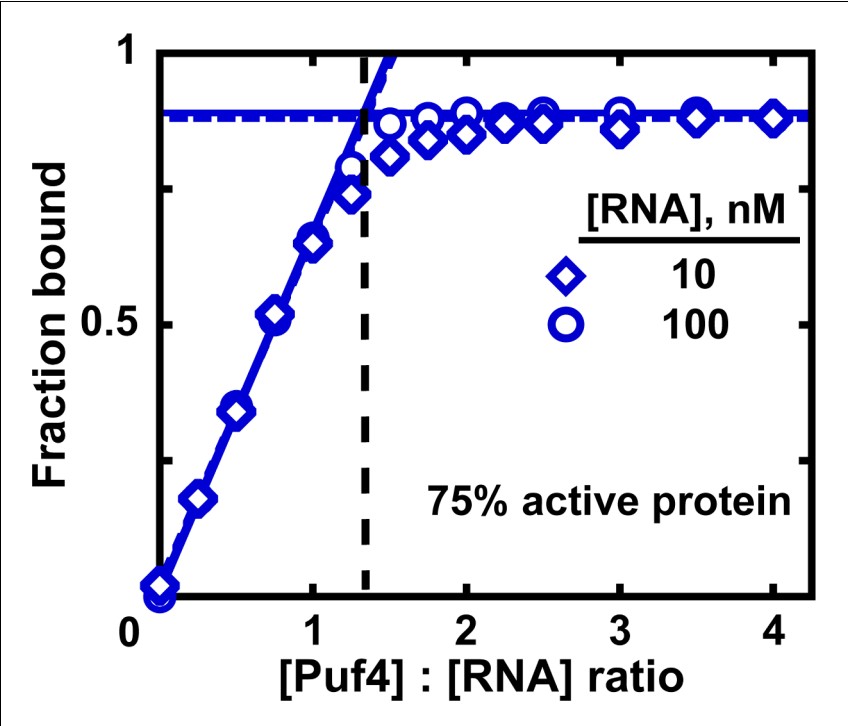

**Figure 7.** Measuring the fraction of active protein by titration. The fraction of active protein is derived from the breakpoint, that is, the intersection of linear fits to the low and high-Puf4 concentration data. See *Figure 7—figure supplement 1* for an alternative strategy using *Equation 5*.

The online version of this article includes the following figure supplement(s) for figure 7:

**Figure supplement 1.** Determination of the fraction of active protein from a quadratic fit.

fold weaker ($k_{off}$ = 1 s$^{-1}$), then no detectable complex is likely to remain after 30 s of washing ($10^{-13}$ of the starting amount). Further, if another RNA ligand binds with the same affinity, but 10-fold slower (and thus also dissociating 10-fold slower; $k_{off}$ = 0.01 s$^{-1}$, half-life of ~100 s), most (~75%) of the complex will remain after the 30 s washing steps despite an identical $K_D$ to the first ligand. In addition, the limited dynamic range of visual readouts of gels that are often used to evaluate pull-down experiments increases the danger of misinterpretation or overinterpretation of these experiments.

Overall, observing binding in pull-downs and related experiments is a complex function of the experimental components and conditions. This doesn't at all mean these experiments should not be done—they often provide critical clues and insights into biology. But, for these and all experiments, we need to keep in mind the nature of the assay, and thus what can and cannot be concluded from the experiment.

Whether binding is absent or not detected can be tested by using approaches that directly report on the equilibrium between bound and unbound components in solution (e.g. ITC, fluorescence anisotropy, and other fluorescence-based techniques), as opposed to indirect approaches like native gel shift and pull-downs that are based on physically separating bound and unbound components, so that unstable complexes may fall apart prior to the detection step. Nevertheless, direct approaches also have limitations. For example, fluorescence intensity or FRET (Förster resonance energy transfer) is limited at high concentrations by inner filter effects, and ITC will miss binding events when the release (or uptake) of heat upon binding is too small (i.e. the binding enthalpy is too small).

A simple way to test whether binding occurs when there is no binding signal is to carry out a competition experiment. If the ligand is bound but not detected in an approach such as native gel shift or filter binding, it will still lessen binding of another ligand for which there is an established signal. The amount lessened depends quantitatively on the $K_D$ values and concentrations of each ligand,

given sufficient time for equilibration. A competition experiment to obtain the $K_D$ value for a weak RNA substrate of Puf4 is shown in Appendix 3, along with the binding scheme and equation to determine the $K_D$ value.

Competition binding measurements can also have a practical benefit; after an initial $K_D$ is determined for a labeled substrate, $K_D$ values for additional substrates can be determined by competition without labeling each substrate (*Hulme and Trevethick, 2010*; *Sanders, 2010*; *Ryder et al., 2008*).

## Discussion

Given the increasingly multi-disciplinary nature of research, scientists are increasingly venturing into disciplines outside their expertise. Our goal is to support these valuable efforts by enabling both experts and non-experts in thermodynamics to get the most out of their binding experiments, and to help them evaluate work by others, published or under review for publication.

While the number of steps described to obtain reliable equilibrium data may initially seem daunting, the accompanying experimental illustrations and guides can transform an opaque process into one that is readily understandable and can be carried out in a straightforward, stepwise fashion by researchers from varied backgrounds.

We found it useful to develop and use an Equilibrium Binding Checklist to organize our approach and findings. We provide a template of such a checklist, along with completed examples in Appendix 4 (*Appendix 4—figure 1*, *2*, *3*). We expect that many readers will find these valuable.

There has been much discussion about problems with reproducibility and rigor in the scientific literature (*Landis et al., 2012*; *Plant et al., 2014*; *Nature, 2013*; *Nosek and Errington, 2017*; *Koroshetz et al., 2020*). Historically, a powerful means to ensure reliability of published data has been to develop community standards. Reporting guidelines have been successfully adopted by journals in a variety of fields, including structural biology (*Berman et al., 2000*), enzymology (http://www.beilstein-institut.de/en/projects/strenda/guidelines), organic synthesis (e.g. http://pubs.acs.org/page/joceah/submission/ccc.html), and many others, and new standards, guidelines and databases are continually being devised (see https://fairsharing.org/ for a curated list). We encourage journals to adopt analogous standards for reporting binding measurements. Contingent on implementation of such standards, we ultimately envision a well-curated and well-documented quantitative database that is routinely used to build and test models for individual molecular interactions and for cellular and molecular networks.

## Materials and methods

### Survey of published equilibrium binding measurements

We surveyed 100 papers, including 66 papers from the list of quantitative RNA/protein studies assembled by the Liu lab (*Yang et al., 2013*) and 34 additional studies reporting $K_D$ and apparent $K_D$ values for RNA/protein interactions (*Supplementary file 1*). To confirm that our survey was not biased, we also scored 20 publications from a single PubMed search for 'RNA protein binding dissociation constant', after confirming that they reported $K_D$ values for RNA/protein binding. Four of the 20 papers also appeared in the above list. The fractions of papers controlling for equilibration and/or titration were similar to those in the main survey (*Figure 1*): 30% of the 20 papers controlled both for equilibration and titration, 15% controlled for neither, 50% only controlled for titration and 5% only controlled for equilibration.

Equilibration was evaluated as follows. If a study reported systematically varying the incubation time, it was counted as controlled for equilibration. If dissociation kinetics were measured in addition to performing equilibrium measurements (n = 3), the study was scored as equilibration-controlled, but only if the reported incubation time was at least three half-lives based on the reported $k_{off}$, and only if the kinetic and equilibrium experiments were performed at the same conditions (n = 1). Studies exclusively using approaches that intrinsically monitor the binding progress (ITC, SPR, biolayer interferometry [BLI]) also were counted as equilibration controlled. However, if several approaches were used in a given study to determine affinities for distinct binding interactions and/or conditions, and if for at least one approach time was not varied, the study was scored as not equilibration

controlled. Some exceptions where equilibration can be reasonably assumed are noted in *Supplementary file 1*.

To generate *Figure 1—figure supplement 1*, we used the incubation times reported for non-equilibration controlled binding experiments. If a narrow range of times (e.g. 15–20 min, 45–60 min; n = 2) was indicated, this was not counted as systematically varying time and the longer time was used for *Figure 1—figure supplement 1*. If only a lower limit of the incubation time was reported (e.g. 'at least 30 min'; n = 1), this lower limit was used for *Figure 1—figure supplement 1*. If two sequential incubations were performed at different temperatures (e.g. '10 min at room temperature and 10 min at 4°C', n = 4), the total incubation time was used for the purposes of the survey. However, since affinity is condition-specific, only equilibration at a constant temperature can yield meaningful $K_D$ values, and two-temperature incubations should be avoided.

To evaluate if titration was controlled for, first, we confirmed if the concentration of the limiting species was systematically varied to determine effects on $K_D$ (n = 5); these studies were counted as titration controlled. If a study reported a range of concentrations of the limiting species, without stating that the effects on $K_D$ were assessed, we did not count this as a titration control, as in practice such a range typically only indicates optimization of radioactive/fluorescent signal to account for radioactive decay and/or varying labeling efficiencies. For the remaining studies, we asked if *Equation 4b* (which assumes the binding regime) or *Equation 5* (which also allows for the intermediate regime) was used to fit the data. If no equation was indicated, or if the midpoint of the binding curve/gel signal was used to determine the $K_D$, or if linear transformation was used in lieu of the hyperbolic fit, we counted the study as using *Equation 4b*. For studies using *Equation 4b*, we asked if the lowest apparent $K_D$ value was in at least 10-fold excess over the limiting component concentration, in which case we counted the study as titration controlled. If a range of limiting component concentrations was reported, we used the lowest value. If only the amount (not concentration) of the limiting species was reported, the concentration was calculated based on the provided volume or, if not indicated, based on a 10 μL reaction volume; nevertheless, binding equilibria depend on concentrations, not amounts, and concentrations, in units of 'M', should always be indicated. If *Equation 5* was used (incl. all ITC measurements), we counted the study as titration controlled, unless the reported $K_D$ was more than 1000-fold below the limiting species concentration (corresponding to a cutoff typically used in ITC [*Velázquez-Campoy et al., 2004*]). For simplicity, we assumed that all SPR/BLI measurements (where the concentration of the immobilized species is difficult to estimate and not reported) were titration controlled; nevertheless, we emphasize the importance of explicitly reporting controls for mass transport in SPR measurements (*Myszka, 1999*). If multiple approaches were used, but at least in one approach titration was not controlled for according to the above criteria, the study was scored as not titration controlled, unless the affected values were corroborated by a titration-controlled approach in the same study.

If no details on the incubation time and/or the concentration of the limiting reagent were provided, but instead a previous study was cited ('as described', n = 4), the information for the above evaluation was obtained from the cited study. This included two cases in which the authors had performed rigorous equilibration and titration controls in their previous referenced work.

## Puf4 purification

The RNA-binding domain (residues 537–888) of *S. cerevisiae* Puf4 was cloned into a custom pET28a-based expression vector in frame with an N-terminal 6X His-tag and a C-terminal SNAP tag (New England Biolabs, Ipswich, MA). The construct was transformed into *E. coli* protein expression strain BL21 (DE3) and protein expression was induced at an OD600 of 0.6 with 1 mM IPTG at 20°C for ~20 hr. Induced cells were harvested by centrifugation at 4500 × g for 20 min. Cell pellets were re-suspended in Buffer A (20 mM HEPES-sodium (HEPES-Na)), pH 7.4, 500 mM potassium acetate (KOAc), 5% glycerol, 0.2% Tween-20, 10 mM imidazole, 2 mM dithiothreitol (DTT), 1 mM phenylmethylsulfonyl fluoride (PMSF) and cOmplete, Mini, protease inhibitor cocktail (Roche Diagnostics GmbH, Mannheim, Germany) and lysed four times using an Emulsiflex (Avestin, Inc, Ottawa, ON, Canada). The lysate was clarified by centrifugation at 20,000 × g for 20 min, nucleic acids were precipitated with polyethylene imine (0.21% final concentration) at 4°C for 30 min with constant stirring and pelleted by centrifugation at 20,000 × g for 20 min. The supernatant was loaded on a Nickel-chelating HisTrap HP column (GE Healthcare, Pittsburg, PA). Bound protein was washed extensively over a shallow 10–25 mM imidazole gradient and eluted over a linear 25–500 mM gradient of imidazole.

Peak Puf4 protein fractions were pooled and desalted into Buffer B (20 mM HEPES-Na, pH 7.4, 50 mM KOAc, 5% glycerol, 0.1% Tween-20, 2 mM DTT) using a desalting column. The His-tag was cleaved by overnight incubation with His-tagged TEV protease at 4°C, and the protein was purified on a HisTrap HP column. The flow-through was desalted into Buffer B and loaded on a HiTrap Q HP column (GE Healthcare) and washed extensively with Buffer B to remove any bound RNA. Protein was eluted over a linear gradient of potassium acetate from 50 to 1000 mM. Protein fractions were pooled and desalted into Buffer C (20 mM HEPES-Na, pH 7.4, 100 mM KOAc, 5% glycerol, 0.1% Tween-20 and 2 mM DTT), concentrated and diluted two-fold with Buffer C containing 80% glycerol for final storage at −20°C. UV absorbance spectra indicated that the protein was free from significant RNA contamination (<1 RNA base per protein).

## RNA 5´-end labeling

Puf4_HO RNA (AUGUGUAUAUUAGU; Integrated DNA Technologies (IDT), Coralville, IA; 5 µM) was labeled with equimolar [γ-$^{32}$P] ATP (Perkin Elmer, Inc, Boston, MA) using T4 polynucleotide kinase (Thermo Fisher Scientific, Vilnius, Lithuania) and purified by non-denaturing gel electrophoresis (20% acrylamide). The RNA was eluted into TE buffer (10 mM Tris-HCl, pH 8.0; 1 mM EDTA) at 4°C overnight, and the lower limit of eluted RNA concentration, assuming no unlabeled RNA, was determined by scintillation counting and calibration against the specific activity of the [γ-$^{32}$P] ATP stock used for labeling. The upper limit of RNA concentration was calculated from total RNA input and the elution buffer volume, assuming a 100% yield.

## Equilibrium binding measurements

All reactions were performed in a binding buffer containing 20 mM HEPES-sodium or HEPES-potassium buffer, pH 7.4, 2 mM magnesium chloride (MgCl$_2$), 100 mM KOAc, 2 mM DTT, 0.2% Tween 20, 5% glycerol, 0.1 mg/ml BSA, at 25 or 0°C, as indicated. The protein and labeled RNA dilutions were prepared in binding buffer at two-times the indicated concentration and were kept on ice until the binding reactions were initiated by mixing 10 µL of protein with 10 µL of labeled RNA. The pipette tips used for mixing and aliquoting the 0°C reactions were kept on ice. The labeled RNA concentrations and incubation times are indicated in the individual figure legends. Following the incubation, 7.5 µL aliquots were moved to 5 µL of ice-cold loading buffer containing 6.25% Ficoll PM 400 (Sigma-Aldrich, Saint Louis, MO), 0.075% bromophenol blue (BPB), and 2.5 µM unlabeled Puf4_HO RNA. The unlabeled RNA in the loading buffer prevented additional association to the labeled RNA from occurring during sample loading (Appendix 2—note 2). Control experiments indicated negligible re-equilibration in loading buffer (t$_{1/2}$ ≥ 3 hr in three independent measurements), consistent with the slow dissociation rate constant measured in binding buffer at 0°C (Appendix 1). All samples were loaded on the gel within 20 min from mixing with the loading buffer. Non-denaturing acrylamide gels (20%) were pre-run for at least 1 hr at 42 V/cm constant voltage, 4–6°C with 0.5x TBE buffer (50 mM Tris, 42 mM boric acid, 0.5 mM EDTA•Na$_2$, pH 8.5–8.6 final) using a circulating cooling system. Aliquots (7.5 µL) were carefully loaded on continuously running gels and separated for 45–90 min. (Extreme caution must be exercised at this step; see, e.g. https://ehs.stanford.edu/reference/electrophoresis-safety for electrical safety hazards.) The gels were dried and exposed to phosphorimager screens, scanned with a Typhoon 9400 Imager and quantified with TotalLab Quant software (TotalLab, Newcastle-Upon-Tyne, UK). Fitting was performed with KaleidaGraph 4.1 (Synergy Software, Reading, PA; RRID:SCR_014980).

The $K_D$ values in *Table 2* indicate the average and standard error from five independent equilibrium experiments (25°C). For 0°C measurements, $K_D$(hyperbolic) indicates the upper limit determined using *Equation 4b* at the lowest RNA concentration (*Figure 6C,D*); $K_D$(quadratic) indicates the average and standard error of $K_D$ values determined with *Equation 5* at the four RNA concentrations shown in *Figure 6C,D*.

## Kinetic measurements

Measurements of $k_{off}$ (Appendix 1) were performed by incubating the indicated concentrations of Puf4 with trace concentration of labeled Puf4_HO RNA for 10 min at 25°C or 0°C in the binding buffer described in *Equilibrium binding measurements*. Labeled RNA concentrations were 0.04–0.5 nM, corresponding to the lower and upper limits, as defined in *RNA 5´-end labeling*. Dissociation

**Table 2.** Summary of equilibrium and kinetic measurements of Puf4 affinity.

| Temperature, °C | Equilibrium* | | Kinetic | | |
| --- | --- | --- | --- | --- | --- |
| | $K_D$(hyperbolic), pM | $K_D$(quadratic), pM | $k_{on}$, $M^{-1}s^{-1}$* | $k_{off}$, $s^{-1}$ | $K_D$ (=$k_{off}/k_{on}$), pM |
| 0 | ≤1.7 | 1.39 ± 0.09 | (2.85 ± 0.14)×$10^7$ | (2.92 ± 0.17)×$10^{-5}$ | 1.02 ± 0.08 |
| 25 | 120 ± 30 | 120 ± 30 | (1.04 ± 0.14)×$10^8$ | 0.014 ± 0.003 | 130 ± 30 |

*The values have been normalized by active protein fraction (75–90%). $K_D$(hyperbolic) and $K_D$(quadratic) refer to values derived from fits to **Equation 4b** and **Equation 5**, respectively. Errors are defined in Materials and methods.

was initiated by transferring the binding reaction to 2.5x volume of unlabeled chase in binding buffer. The chase RNA concentrations in the final reaction were 250 nM and 1000 nM. At various times, 7.5 µL aliquots were moved to 5 µL of ice-cold loading buffer containing 6.25% Ficoll PM 400% and 0.075% BPB, and 7.5 µL aliquots were loaded on a pre-run, continuously running 20% non-denaturing gel at 4–6°C. All pipette tip boxes and solutions used for the 0°C reactions were kept on ice. The chase solution for the 25°C reaction was pre-warmed in a 25°C water bath for 10 min before initiating the dissociation reaction. All time courses were fit to single exponentials using KaleidaGraph 4.1.

The effectiveness of unlabeled Puf4_HO RNA chase was tested by pre-incubating 10 nM Puf4 with 100–1000 nM unlabeled RNA (final concentrations) for 12 min at 25°C before adding trace amount of labeled Puf4_HO RNA (0.04–0.4 nM). The fractions of bound labeled RNA ranged from 0.01 (1000 nM) to 0.1 (100 nM), compared to 0.95 fraction bound in the absence of chase, confirming the effectiveness of the chase.

The $k_{off}$ values reported in *Table 2* indicate the average and standard error from two replicate experiments (25°C) or the average and standard error across different concentrations in a single experiment (0°C).

Values of $k_{on}$ were determined by mixing 40 µL each of trace labeled RNA solution (0.004–0.05 nM) and varying dilutions of Puf4. At varying times, 7.5 µL aliquots were transferred to 5 µL of ice-cold loading buffer containing 6.25% Ficoll PM 400, 0.075% BPB, and 2.5 µM unlabeled Puf4_HO RNA and loaded on a 20% gel as above. The protein and RNA solutions were pre-incubated at the reaction temperature (0°C or 25°C) before mixing, and ice-cold tips were used for the 0°C reactions. To control for titration by labeled RNA at the low protein concentrations used, at 0°C, the equilibration rate constants were also measured at three-fold higher labeled RNA concentration, giving consistent rate constants within 1.1–1.3-fold (Appendix 1).

The $k_{on}$ values reported in *Table 2* are the slopes and standard errors of linear fits to observed rate constants from two replicate experiments (25°C) or a single experiment (0°C). The $k_{on}$ values were corrected for the active protein fraction.

## Measuring the fraction of active protein by titration

Unlabeled Puf4_HO RNA (10 or 100 nM) was incubated for 30 min with varying Puf4 concentrations in the presence of trace labeled Puf4_HO RNA (0.06–0.4 nM); the labeled and unlabeled RNA was pre-mixed before adding Puf4. The fraction bound RNA was determined as described in *Equilibrium binding measurements*.

## Competition measurements

Trace labeled Puf4_HO RNA (0.02–0.19 nM) was equilibrated with 0.4 nM or 1.2 nM Puf4 and diluted two-fold into solutions containing varying concentrations of unlabeled competitor RNA (CGUAUAUUA; IDT). The reactions were incubated at 25°C for the indicated time, followed by transfer of 7.5 µL aliquots to 5 µL ice-cold loading buffer (6.25% Ficoll PM 400, 0.075% BPB, and 2.5 µM unlabeled Puf4_HO RNA). The samples were loaded immediately on a continuously running native acrylamide gel (4–5°C). The curves were fit to *Equation 9*, as described in Appendix 3.

## Simulations

The simulated data in *Figure 5* were generated by using *Equation 4b* (panel A) and *Equation 5* (panel B) to calculate the fraction of bound RNA at each total protein concentration. In *Figure 5—*

*figure supplements 1*, *2*, *4* and *5*, *Equation 5* was used to calculate fractions bound at each protein and ligand concentration. In *Figure 4—figure supplement 1*, *Equation 4b* was used to determine the fraction of ligand bound at each protein concentration at equilibrium, assuming $[P] = [P]_{total}$. This equilibrium value was then used as an amplitude (A) term in the single-exponential equation shown in *Figure 2* to determine the fraction of bound ligand at each time point t: Fraction bound(t) = $A \times (1 - e^{-t \times k_{equil}}) = $ Fraction bound(equilibrium) $\times (1 - e^{-t \times (k_{on}[P] + k_{off})})$.

The simulated data in *Figure 5—figure supplement 3* were generated as follows. First, *Equation 5* was used to calculate the expected fraction of bound RNA at equilibrium for each $[R]_{total}$ and $[P]_{total}$ indicated in the figure. Two-fold serial dilution of protein was chosen as representative of a typical equilibrium binding experiment. In the case of 0.001 nM $R_{total}$, *Equation 4b* was used instead to calculate the expected fraction bound, as this condition satisfies the $[P]_{free} = [P]_{total}$ assumption. Random noise in fraction bound was then generated around each predicted data point by sampling from a normal distribution with the indicated standard deviation, using the scipy and random packages in Python. Ten binding series were generated this way for each condition and each noise level. These datasets were then individually fit to *Equation 5* (or *Equation 4b* in the case of 0.001 nM $R_{total}$) in Prism 8 (GraphPad Software, LLC, San Diego, CA; RRID:SCR_002798), with the equations modified to include amplitude (A) and y axis offset (O) terms:

$$\text{Fraction bound} = A \times \frac{([R]_{total} + [P]_{total} + K_D) - \sqrt{([R]_{total} + [P]_{total} + K_D)^2 - 4 \times [R]_{total} \times [P]_{total}}}{2 \times [R]_{total}} + O \qquad (6)$$

$$\text{Fraction bound} = A \times \frac{[P]_{total}}{[P]_{total} + K_D} + O \qquad (7)$$

To facilitate fitting to *Equation 6*, $[R]_{total}$ was constrained to the known value, and the $K_D$ was constrained to positive values only, with the real affinity (0.1 nM) used as an initial estimate.

## Acknowledgements

We thank Geeta Narlikar, Enrique De La Cruz, and members of the Herschlag lab for discussions and comments. We are also grateful to Hashim Al-Hashimi, Tom Cech, Katrin Karbstein, Olke Uhlenbeck, Chris Walsh, and Deborah Wuttke for critical feedback and suggestions. This work was funded by a grant from the US National Institutes of Health to DH (R01 GM132899).

## Additional information

### Funding

| Funder | Grant reference number | Author |
| --- | --- | --- |
| National Institutes of Health | R01 GM132899 | Daniel Herschlag |

The funders had no role in study design, data collection and interpretation, or the decision to submit the work for publication.

### Author contributions

Inga Jarmoskaite, Conceptualization, Data curation, Formal analysis, Validation, Investigation, Visualization, Methodology, Writing - original draft, Writing - review and editing; Ishraq AlSadhan, Formal analysis, Validation, Investigation, Visualization, Writing - review and editing; Pavanapuresan P Vaidyanathan, Conceptualization, Resources, Data curation, Visualization, Methodology, Writing - original draft, Writing - review and editing; Daniel Herschlag, Conceptualization, Supervision, Funding acquisition, Writing - original draft, Project administration, Writing - review and editing

## Author ORCIDs

Inga Jarmoskaite (iD) https://orcid.org/0000-0001-5847-5867
Daniel Herschlag (iD) https://orcid.org/0000-0002-4685-1973

## Decision letter and Author response

Decision letter https://doi.org/10.7554/eLife.57264.sa1
Author response https://doi.org/10.7554/eLife.57264.sa2

## Additional files

### Supplementary files

- Supplementary file 1. Literature survey of 100 RNA/protein binding studies.
- Supplementary file 2. Literature survey of CRISPR nuclease binding studies, representative of high-affinity interactions.
- Transparent reporting form

### Data availability

No datasets were generated in this work. The figures include all data, or, where most appropriate for clarity, representative data from a single experiment for every type of experiment performed.

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

## Appendix 1

### Kinetic approach to affinity determination

An equilibrium dissociation constant is the ratio of dissociation and binding rate constants ($K_D = \frac{k_{off}}{k_{on}}$), and thus can be determined by directly measuring these rate constants. Because $k_{off}$ is concentration-independent, it is the easiest and most robust parameter to measure. *Appendix 1—figure 1* describes the steps for this measurement. After forming the complex between protein and a trace concentration of labeled RNA, a large excess of unlabeled RNA is added to the reaction. The role of the unlabeled 'chase' RNA is to bind any dissociated protein before it can re-bind the labeled RNA. Thus, the chase RNA must be in large excess of the protein concentration and must be a tight binder. The probability of rebinding can be further reduced by diluting the reaction mixture. At specified time points ($t_2$; *Appendix 1—figure 1A*), the amount of remaining complex can be determined by native gel electrophoresis or another approach. Although in principle $k_{off}$ can be determined from a single binding reaction, as in any experiment, reliability is best established with several controls (see Appendix 2—note 6).

*Appendix 1—figures 1B, C* show dissociation of RNA from Puf4 at 25°C and 0°C, respectively. As expected for simple dissociation with an effective chase, the curves are well fit by a single exponential curve with endpoints that approach zero and the rate constant is independent of protein and chase concentrations. A critical control is to test that the $k_{off}$ is not affected by the chase concentration. This is because in some contexts of multi-step dissociation processes, the chase can facilitate dissociation (e.g. *Hadizadeh et al., 2016*).

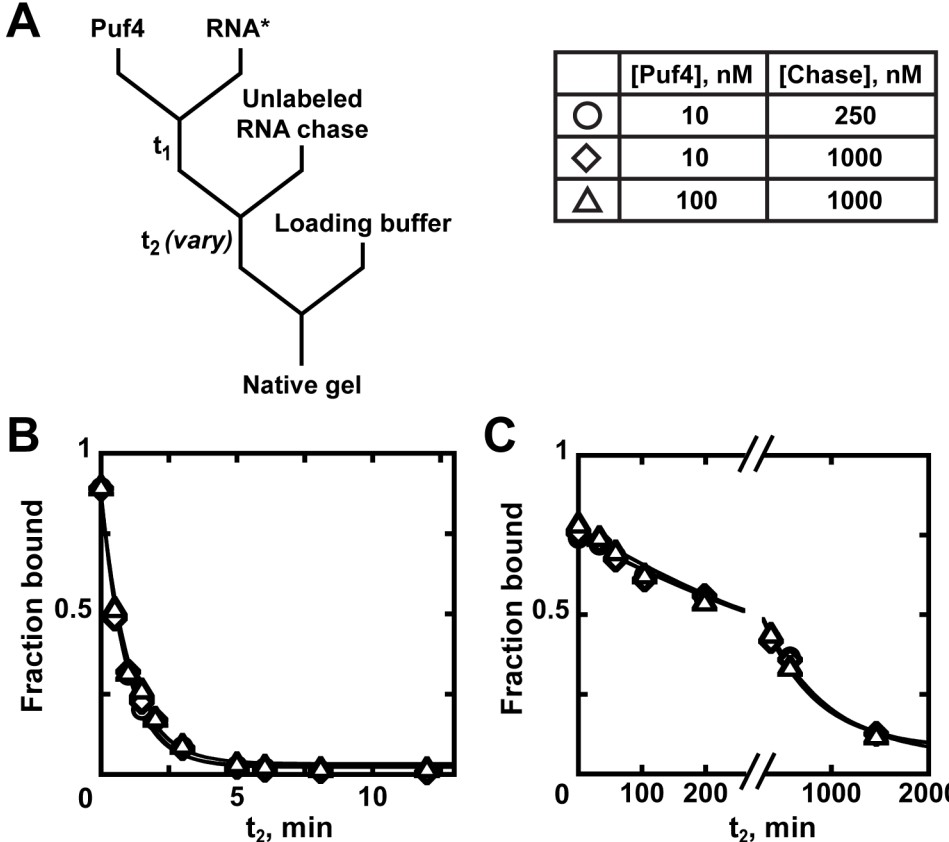

**Appendix 1—figure 1.** Kinetics of Puf4 dissociation. (**A**) Mixing scheme for measuring the dissociation rate constant. After equilibration of a saturating or near-saturating concentration of Puf4 protein with a trace concentration of labeled RNA ($t_1$), a large excess of unlabeled RNA is added, with concomitant dilution of the binding reaction to prevent rebinding after dissociation. (**B–C**) Time dependence of Puf4 dissociation from its consensus RNA at 25°C (C; $k_{off} = (0.014 \pm 0.003)$ s$^{-1}$) and at 0°C (D; $k_{off} = (2.92 \pm 0.17) \times 10^{-5}$ s$^{-1}$).

To measure the association rate constant one can use a different type of chase experiment that we refer to as a '$k_{on}$ chase' (**Appendix 1—figure 2A**; **Hertel et al., 1994**). Here the time that the protein and labeled RNA are incubated together is varied ($t_1$) and the amount bound after each time $t_1$ is determined by native gel shift or another assay. To ensure that the amount bound accurately reflects what has occurred during $t_1$ and not subsequently, a chase is added to prevent, or quench, additional binding, analogous to the $k_{off}$ experiment above (**Appendix 1—figure 1A**). The time $t_2$ is kept constant, removing potential variability from dissociation subsequent to the binding reaction during $t_1$ (see Appendix 2—note 2).

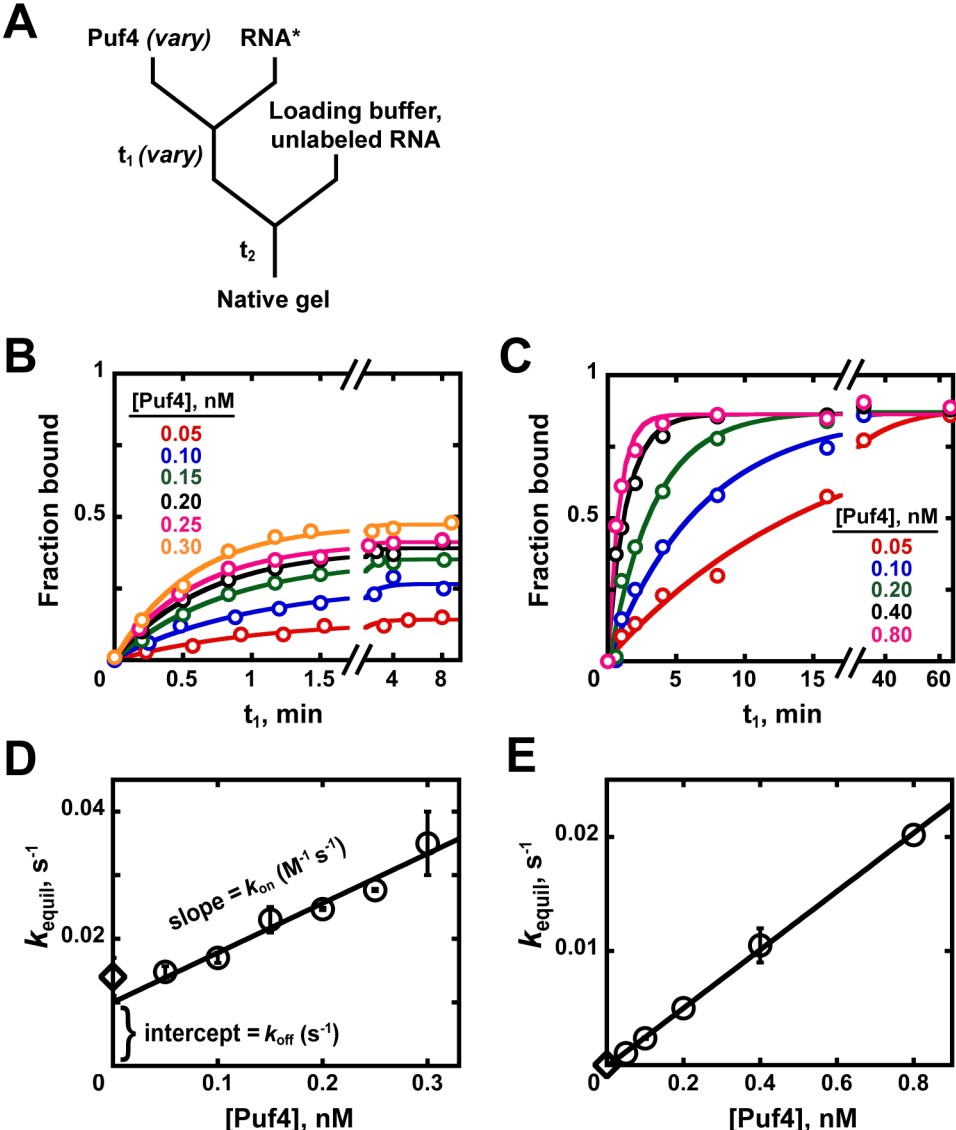

**Appendix 1—figure 2.** Kinetics of Puf4/RNA association. (**A**) Mixing scheme for measuring association rate constants. (**B, C**) Time dependence of Puf4 association to its consensus RNA at 25°C (**B**) and 0°C (**C**). (**D, E**) Determination of $k_{on}$ from the slope of the Puf4 concentration dependence of equilibration rate constants in parts B and C, respectively (circles). The $k_{off}$ values from **Appendix 1—figure 1** are also shown (diamonds) to illustrate the correspondence between the y-intercept and $k_{off}$ (**Equation 1**). Panels D and E show results from two and one independent experiments, respectively (error bars in $E$ correspond to averages from measurements at two different labeled RNA concentrations).

The observed association rate constant is expected to vary with protein concentration—that is, it is first order in protein (**Figure 3**)—so it is important to carry out these measurements across a wide

range of protein concentrations. *Appendix 1—figures 2B, C* show the data obtained at 25°C and 0°C, respectively. Each individual time course is well fit by an exponential, and *Appendix 1—figures 2D, E* plot the rate constants obtained from these time courses versus Puf4 concentration, giving the expected linear dependencies, the slopes of which correspond to $k_{on}$ (*Appendix 1—figure 2D*).

The plot in *Appendix 1—figure 2D* also shows a clear, non-zero intercept. While not intuitive, this intercept arises because the '$k_{on}$' experiment actually measures the rate constant to reach equilibrium, $k_{equil}$, where $k_{equil}$ equals $k_{on}[P] + k_{off}$ (*Equation 1*) so that the slope gives $k_{on}$ and the intercept gives $k_{off}$ (*Appendix 1—figure 2D*). There is good agreement between the intercepts and the independently measured $k_{off}$ values in our experiments (*Appendix 1—figures 2D, E*, diamonds). It is generally preferable to compare directly-obtained $k_{off}$ values to these intercepts, rather than relying on the intercept for $k_{off}$ determination, as this allows independent tests of data consistency and accuracy.

The $K_D$ values obtained in the equilibrium and kinetics experiments agree within two-fold, which is reasonable experimental agreement in our experience (*Table 2*). Such agreement strongly supports (although does not prove) that both methods are giving correct binding constants.

## Appendix 2

### Additional considerations

#### Note 1: $k_{off}$ as shortcut for establishing equilibration times

Measuring $k_{off}$ provides a fast and dependable way to determine the equilibration time needed for simple two-state binding reactions (*Figure 3* and *Equation 2*). However, we still recommend monitoring the time course of complex formation in the presence of ligand, in case binding is more complex than a single step, for example involving an additional slow conformational step (e.g. *LeCuyer and Crothers, 1994*; *Smith et al., 2009*; *Bevilacqua et al., 1992*; *Mueller-Planitz and Herschlag, 2008*; *James and Tawfik, 2005*; *Pisareva et al., 2006*). If there is a slow step preceding binding, the rate of equilibration may become limited by this slow step. For example, the formation of long-lived stable alternative structures is well known for RNA (e.g. *Uhlenbeck, 1995*; *Herschlag, 1995*). Such alternative states can lead to rapid equilibrium binding for a sub-population and then slow binding as the misfolded, alternative state re-equilibrates to give partial binding or non-exponential kinetics. The following are diagnostics for these and related issues:

I. Association and/or dissociation kinetics do not follow a single exponential. Such more complex kinetics indicate the presence of additional species that must be identified.

II. Association kinetics are not first order in protein; that is, the binding rate constant is independent of protein concentration instead of the linear dependence seen in *Appendix 1—figure 2D, E* and predicted by *Equation 1*. This behavior indicates additional species in the binding process.

III. Equilibration rate constant *is* dependent on protein concentration, but binding does not go to completion even at saturating concentrations. E.g. if only half of a ligand is bound at saturation, that may indicate that 50% of the ligand is trapped in a long-lived conformation or covalently heterogeneous (e.g. protein ligands that are partially or heterogeneously covalently modified). Alternatively, incomplete binding can mean that the ligand dissociates during disruptive sample processing steps (see Note 2). Further tests, such as extending the incubation time, pre-incubating an RNA ligand at increased temperatures, or analysis of the labeled ligand by HPLC, gel, sedimentation, or other methods, can determine if incomplete binding is caused by a slow conformational step or by covalent heterogeneity.

If these situations do not apply, $k_{off}$ is sufficient to determine the required equilibration time.

#### Note 2: Controls for changes during sample processing in 'indirect' binding measurements, and approaches to prevent these changes

Techniques such as native gel shift, nitrocellulose filter binding, and any pull-down-based approaches (*Lambert et al., 2014*; *Wong and Lohman, 1993*; *Ryder et al., 2008*; *Campbell et al., 2012*) involve sample processing steps between the binding incubation and detection of bound complex, and are thus 'indirect.' Changes in conditions during sample handling and analysis can perturb the amount of complex from that present at the end of the initial incubation. It is therefore important to control for and ideally prevent such changes, and additional confirmatory experiments, such as the kinetic experiments described in the main text, are necessary to obtain high-confidence $K_D$ values.

In the case of native gel shift experiments, following the incubation (time $t_1$ in *Figure 4A*), the sample is transferred to a gel loading buffer, containing a high-density additive such as Ficoll or glycerol to facilitate gel loading, and typically a dye, and is loaded onto a non-denaturing gel. These steps are accompanied by changes in concentration, solution conditions, and often temperature, all of which have the potential to induce dissociation or additional binding.

Some general strategies to minimize such perturbations include:

I. Minimizing any changes in conditions between the incubation ($t_1$) and detection ($t_2$) steps. E. g. the loading buffer can be omitted altogether by including sufficient glycerol in the reaction itself and the gel can be run at the same temperature as the binding reaction (*Hellman and Fried, 2007*). However, this will not always be feasible, e.g. due to rapid dissociation of the complex during room-temperature gel electrophoresis.

II.   Empirically assessing the effects of any changes in conditions on binding and the time scales on which these effects occur. Certain changes will have negligible and consistent effects on binding and will not affect the quantification if samples are handled quickly and consistently.

III.  Varying the original incubation conditions while maintaining the same gel loading and gel running conditions is a worthwhile control to establish that the observed fraction bound does reflect at least some property from the original incubation conditions.

For our Puf4 binding assays, we were able to prevent changes during the native gel shift assay by utilizing certain favorable properties of Puf4/RNA binding. Below we describe the specific steps we took, as some of the Puf4 strategies can be adapted to other systems with similar properties.

1.  Preventing complex dissociation:
    ▪  During initial exploration, we found that Puf4 dissociates from its consensus RNA extremely slowly at 0°C (Appendix 1). Thus, by keeping our loading buffer on ice and running the gels at 4–5°C we were able to effectively 'quench' complex dissociation, with only negligible dissociation ($t_{1/2} \geq 3$ hr) occurring during the short time (seconds–minutes) the samples spent in loading buffer or the gel running buffer.
    ▪  While dissociation of weaker Puf4 ligands was non-negligible even at low temperatures (data not shown), loading these samples quickly (within seconds) and with a consistent loading time ensured that dissociation only affected the amplitude and not the shape of the equilibrium binding curve (and thus the $K_D$ determined from it). This was confirmed by competition measurements.

2.  Preventing additional complex formation:
    ▪  In contrast to the slow dissociation, Puf4 association rate constant remains high even at 0°C (Appendix 1). Thus, additional binding can occur during sample loading, and the amount of additional binding will vary with the concentrations of the binding partners and the time prior to loading and gel entry.
    ▪  To avoid the above complexities, we included a large, saturating excess of unlabeled RNA in the loading buffer (here we used the same oligonucleotide as the labeled RNA; more generally—a tight binder that binds at least as tightly as the labeled RNA should be used). This is equivalent to the $k_{on}$ chase used in the kinetics measurements (Appendix 1) and ensures that the additional binding that occurs before entering the gel is to the unlabeled RNA—such that the fraction of bound labeled RNA still accurately reflects the fraction bound during the original incubation.
    ▪  If applying an analogous chase approach, it is important to keep in mind that the unlabeled RNA concentration in the loading buffer must be at least 10-fold higher than the protein concentration used. Otherwise, a substantial fraction of labeled RNA can still bind during $t_2$, in a manner dependent on protein concentration. Using a chase RNA that is bound more tightly than the ligand being tested (e.g. wild-type RNA sequence vs. a mutant) allows the use of lower excess.

Additional measures to minimize changes during binding measurements by native gel shift:

I.    To maintain consistent loading time for all samples, and to minimize the time in the gel running buffer, the samples should be (carefully!) loaded on a continuously running gel. (DANGER: As there is sufficient current to cause injury or death, extreme caution is required in this step. Always be sure there are no leaks, never touch the gel while running, even with gloves on, and maintain a safe distance as current can arc; see, e.g. https://ehs.stanford.edu/reference/electrophoresis-safety for safety information.)

II.   The ratio of the sample volume to the area of the bottom of the well should be kept as low as possible. This ratio can be optimized by loading different sample volumes and varying the comb size and the gel thickness.

III.  The percentage of acrylamide is another variable that should be adjusted if excessive dissociation on the gel occurs (indicated by smearing), with higher acrylamide percentages recommended to increase complex stability in the gel (see also *Altschuler et al., 2013*).

IV.   Using high-density compounds such as Ficoll or glycerol in the loading buffer facilitates rapid gel entry by concentrating the sample at the bottom of the well.

V.    It is advisable to vary the above and other factors (including voltage and temperature) to determine if they influence the results. Factors following sample incubation should not affect the results.

## Note 3: Number of time points for establishing equilibration time

Although in principle using two well-separated equilibration times is sufficient, using three or more times is preferable. Using only two incubation times has the potential to give a misleading result—if binding continues to increase while, for example, protein is denaturing, these factors can cancel each other out to give apparently constant binding. It is also critical to use times that span a considerable range, preferably approaching or exceeding 10-fold; here the concern is that if the time interval is narrow, the (inevitable) measurement error can make it difficult to distinguish if measurements are or are not time-independent.

## Note 4: Uncertainty in the concentration of labeled trace binding partner

Accurately quantifying trace concentrations of labeled ligands can be a challenge, for instance, when working with radioactively labeled oligonucleotides. The concentration of labeled material can be estimated from specific activity of the isotope used for labeling. However, if labeling is incomplete and the purification procedure used after labeling does not fully separate labeled from unlabeled material, more oligonucleotide will be present than accounted for by specific activity. To be conservative, the upper limit of labeled oligonucleotide concentration should always be calculated based on the total oligonucleotide input in the labeling reaction. The concentration determined by specific activity provides a lower limit.

If not all protein is active or if the protein concentration is inaccurate, one may have false confidence of being in the excess-protein regime described by *Equation 4b*. Thus, although keeping the trace component at least 10-fold below the dissociation constant is a useful benchmark, we recommend always varying the trace ligand concentration (section 'Avoid the titration regime'). When possible, the concentration of active protein should be determined by titration against a known concentration of the ligand (see section 'Determine the fraction of active protein').

## Note 5: Additional factors that can affect binding-competent concentrations of protein and ligand

Depending on the system and technique, other factors can lead to concentration-dependent variation in the observed $K_D$ values. Performing the simple controls in the Equilibrium Binding Checklist (Appendix 4) will typically detect these problems and help assess the robustness of measured $K_D$ values. We provide a non-exhaustive list below that will be helpful in devising appropriate additional controls, but experimenters should consult references for technique-specific information and advice.

I. Proteins can denature over time, and can also be subject to time-dependent proteolysis in extracts or partially purified systems.
   - Apparent weakening of binding with increased incubation time will help detect this behavior (for reactions where protein is the excess binding partner).
   - Optimizing solution conditions (e.g. lowering temperature, including glycerol, varying pH, etc.) and using protease inhibitors may help extend accessible times.
   - $k_{off}$ measurements can help identify the shortest feasible time to be used for equilibrium incubation to limit damage (in cases where damage occurs during unnecessarily long incubations).
II. Proteins can aggregate/form higher-order complexes at very high concentrations.
   - Active protein concentration (section 'Determine the fraction of active protein') should be assessed at several protein concentrations, including one at the high-end of the concentration range used in equilibrium experiments (see also discussion in *Altschuler et al., 2013*).
   - If concentration-dependent changes in stoichiometry of the complex are detected (which can be detected with some approaches like gel electrophoresis, certain fluorescence-based methods), models beyond the simple model in *Figure 3* should be devised and tested.
III. Single-stranded oligonucleotides can form intermolecular base-pairs when used at high concentrations (e.g. in competition experiments) or during storage.
   - Nearest-neighbor base-pairing predictions can be used to estimate if base-pairing may be an issue at the temperature, salt conditions, and oligonucleotide concentrations used. If interactions between oligonucleotides are suspected, one can weaken base-pairing by using higher temperature or lowering salt concentrations in the binding experiments. If

 such changes are not possible or sufficient, the binding model should be modified to incorporate the oligonucleotide interactions.

- For nucleic acid constructs with extensive complementarity that can form long-lived intermolecular interactions during storage, a dilute solution should be heated before use in binding experiments.

IV. Nucleic acids can be covalently damaged by nucleases and other factors.

- Care should be taken to remove all potential nuclease contamination by using sterile, high-purity water and reagents, sterile supplies and surface decontaminants. In our experience, some of the most robust RNase contamination has come from contaminated lots of commercial RNase *inhibitors*; thus, we recommend testing these products before relying on their efficacy.
- UV exposure should be limited or avoided during nucleic acid purification to avoid covalent damage (*Kladwang et al., 2012*; *Greenfeld et al., 2011*).

V. Both proteins and nucleic acids can stick to tubes, lowering the concentrations accessible for binding.

- Varying the type of reaction tube, and including small amounts of detergent and bovine serum albumin (BSA) can be used to assess and prevent sticking. (NOTE: Some BSA and other protein preparations contain nuclease contaminants.)
- Varying the concentration of the labeled trace partner and measuring the dissociation constant by more than one approach (equilibrium vs. kinetics, or different techniques) can control for loss of material due to sticking.

VI. Long-lived misfolded RNA concentrations can reduce binding-competent concentration during short incubation times (see Note 1).

## Note 6: Controls and considerations for dissociation rate constant measurements

We recommend the following steps to ensure accurate $k_{off}$ measurements.

I. *Establish that the chase is effective.* Mixing the chase (in large excess over the protein concentration) with labeled ligand before addition of protein to form the complex should lead to no detectable protein binding to labeled ligand. If this is not the case, a higher chase concentration and/or a higher-affinity chase ligand is needed.

II. *Establish independence of* $k_{off}$ *from the chase ligand concentration.* Multiple chase ligand concentrations should be used, preferably spanning at least an order of magnitude. It is expected that $k_{off}$ will be constant, but variation can indicate an experimental artifact (such as a chase component affecting $k_{off}$) or, more interestingly, an ability of one ligand to facilitate dissociation of another. Here, dissociation by dilution becomes an important control— i.e. the bound complex formed with protein concentration near the $K_D$ is diluted by varying amounts until full or near-full dissociation is observed. The dissociation rate constant should be consistent across different dilution factors that give full dissociation. Incomplete dissociation (with the dissociation curve plateauing substantially above zero) most simply suggests insufficient chase (and/or dilution), which is usually readily resolved by increasing the chase concentration. Less commonly, incomplete dissociation can indicate heterogeneity of the bound complex, with a slowly dissociating sub-population remaining bound on the time scale of the experiment. In this case, increasing the chase concentration will not lead to complete dissociation, and the origins of complex heterogeneity should be investigated. Indeed, the more slowly dissociating fraction is more likely to represent a functional form, as it is more tightly associated.

III. *Establish independence of protein concentration.* While the starting fraction bound may vary depending on how far above the $K_D$ the protein concentration is, the dissociation rate constant should always be the same. Changes in $k_{off}$ with protein concentration can indicate a contaminant in the protein solution or, more interestingly, the formation of a protein multimer that increases or decreases the RNA dissociation rate. Thus, $k_{off}$ measurements at several protein concentrations (ideally three or more spanning at least an order of magnitude), in addition to serving as controls, can help discover new complexes and pathways.

## Appendix 3

### Competition measurements

Considerations for competition measurements

I.  Protein concentrations ~2–5 times above the $K_D$ for the labeled ligand ($K_D^*$ in **Appendix 3—figure 1A**) should typically be used. It is important to be near or above the $K_D^*$ to have sufficient signal, but at the same time not too far above the $K_D^*$ so that the protein concentration is sufficiently below the $K_D$ of unlabeled competitor ($K_{D,comp}$) for most of the competitor to remain unbound (analogous to the 'trace' condition recommended for direct binding measurements; see section 'Avoid the titration regime'). Under these conditions ($[P]_{total} \ll K_{D,comp}$), the quadratic solution to the 'Lin and Riggs' equation can be used to determine $K_{D,comp}$ (**Equation 9**, **Appendix 3—figure 1C**; **Lin and Riggs, 1972**). Protein concentrations below $K_D^*$ can be used as controls for competitive binding (analogous to varying the labeled ligand concentration to rule out titration in direct binding measurements).

II.  If the unlabeled competitor binds with a $K_D$ very similar to, or lower than that of the labeled ligand, the condition $[P]_{total} \ll K_{D,comp}$ for **Equation 9** will not be satisfied. In such case, the cubic equation, which accounts for the depletion of the competitor, can be used (**Wang, 1995**). For the high-affinity competitors, it may be more reliable to measure binding directly (rather than by competition); alternatively, a higher-affinity labeled species could be used.

III.  The competitive binding equation includes labeled ligand concentration (**Appendix 3—figure 1C**). In a typical experiment where labeled ligand is used in trace ($[R^*]_{total} \ll K_D^*$ in **Appendix 3—figure 1C**) this value is negligible and does not contribute substantially to the $K_{D,comp}$. Nevertheless, it is still important to establish whether the labeled ligand concentration affects $K_{D,comp}$ in a given experiment by fitting the data with lower and upper limits of labeled ligand concentration used in the equation (see Appendix 2—note 4). In case of the Puf4 experiment, using the upper and lower limits of labeled RNA concentration resulted in similar $K_{D,comp}$ values (230 nM and 204 nM, respectively), but using the upper limit led to a poorly fit amplitude due to expected slight protein depletion. This observation, along with others, suggests that the real labeled RNA concentration was more closely approximated by the lower limit (based on $^{32}$P quantification) than the upper limit (based on total RNA used for R* preparation), and that negligible co-purified unlabeled RNA was present in our R* preparation (Appendix 2—note 4).

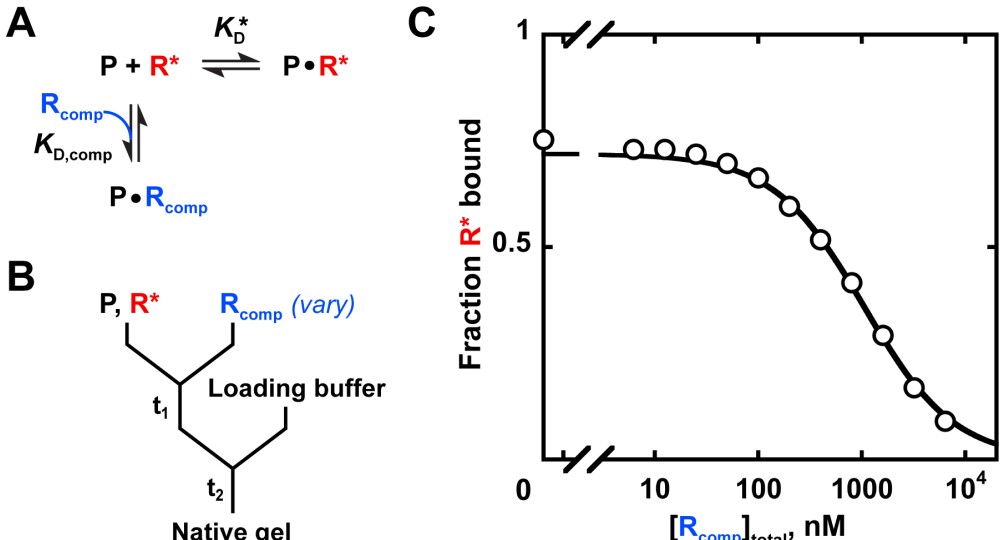

**Appendix 3—figure 1.** Measuring binding affinity by competition. (**A**) Competitive binding reaction scheme. R*: labeled RNA ligand; $R_{comp}$: unlabeled competitor RNA; $K_D^*$: protein affinity for R*; $K_{D,comp}$: protein affinity for $R_{comp}$. (**B**) Mixing scheme for a competition measurement. (**C**)

Competition between the U1C point mutant of the Puf4 consensus ($R_{comp}$ = CGUAUAUUA) and the labeled consensus RNA ($R^*$= $^{32}$P-AUGUGUAUAUUAGU). The data were fit to the following equation (*Lin and Riggs, 1972*; *Weeks and Crothers, 1992*):

$$\text{Fraction bound} = A \times \frac{1}{2[R^*]_{\text{total}}} \left[ K_D^* + \frac{K_D^*}{K_{D,\text{comp}}}[R_{\text{comp}}]_{\text{total}} + [P]_{\text{total}} + [R^*]_{\text{total}} - \sqrt{\left( K_D^* + \frac{K_D^*}{K_{D,\text{comp}}}[R_{\text{comp}}]_{\text{total}} + [P]_{\text{total}} + [R^*]_{\text{total}} \right)^2 - 4[R^*]_{\text{total}}[P]_{\text{total}}} \right] + O \quad (9)$$

A indicates the maximum amplitude, constrained to the fit amplitude of the R* binding curve that is measured in parallel by a direct binding experiment (A = 0.89 for Puf4 binding to R*). O is the y axis offset (background). $[R^*]_{\text{total}}$ was constrained to the lower limit of the labeled RNA concentration. $K_D^*$ was constrained to Puf4 affinity for the labeled RNA, as determined by direct measurement in the same experiment (0.105 nM, after accounting for active protein fraction of 75%). $[P]_{\text{total}}$ was 0.45 nM, after accounting for active protein fraction. The fit $K_{D,\text{comp}}$ value was 204 nM. Incubation times of 10, 30, and 110 min gave consistent $K_{D,\text{comp}}$ values (190–210 nM), as did lowering the protein concentration by three-fold (180 nM). *Equation 9* is applicable only for $K_{D,\text{comp}} >> K_D^*$. For other cases see *Wang, 1995*.

## Appendix 4

### Equilibrium binding checklist

---

**Equilibrium Binding Checklist**

**Binding partner 1 (BP1):** _______________________________________________________

**Binding partner 2 (BP2):** _______________________________________________________

**Method:** _______________________________________________________

**CONDITIONS:**   **Temperature:**________________   **Buffer & pH:**________________________________

   **Salt(s):**________________________________________________________

   **Other:** _________________________________________________________

---

**A. Required:**

☐  1. Vary incubation time to test for equilibration.

   Time range:____________________   Number of time points:________________

   BP1 concentration(s):________________   BP2 concentration(s):________________

   Time-independence across the entire binding curve?   Y ☐   N ☐

☐  1.1. *Alternative approach*: measure $k_{off}$.

   $k_{off}$:____________________   Calculated equilibration time (5 half-lives): ________________

☐  2. Vary the concentration of both binding partners.

   Concentration range of 'trace' binding partner:________________________________

   $K_D^{app}$ independent of trace binder concentration?   Y ☐   N ☐

   Concentration range showing invariant $K_D^{app}$ :________________________________

   Binding equation used:      ☐ hyperbolic      ☐ quadratic

   Binding curves shown?   Y ☐   N ☐

   Systematic deviations from the binding curve?   Y ☐   N ☐

   $\boldsymbol{K_D^{app}}$ :____________________ *(upper limit if dependent on trace binder concentration)*

---

**B. Recommended:**

☐  1. Test $K_D$ by an independent approach.

   Alternative approach: ________________________________________________

   $K_D^{app}$ from alternative approach:________________

☐  2.  Determine the fraction of active protein by titration.

   $K_D$ corrected for active protein fraction?   Y ☐   N ☐

   Fraction of active protein:____________________

---

**Comments:**

---

**Appendix 4—figure 1.** Equilibrium binding checklist template.

**Equilibrium Binding Checklist**

**Binding partner 1 (BP1):** PUF domain of *S. cerevisiae* Puf4 with C-terminal SNAP tag

**Binding partner 2 (BP2):** [5'-$^{32}$P]-AUGUGUAUAUUAGU RNA

**Method:** native gel shift

**CONDITIONS:** **Temperature:** 25°C **Buffer & pH:** 20 mM HEPES-K/Na, pH 7.4

 **Salt(s):** 2 mM MgCl$_2$, 100 mM KOAc

 **Other:** 2 mM DTT, 0.2% Tween 20, 5% glycerol, 0.1 mg/ml BSA

---

**A. Required:**

☑ 1. Vary incubation time to test for equilibration.

 Time range: 0.5–4.5 hr Number of time points: 3

 BP1 concentration(s): 0.0036–205 nM BP2 concentration(s): 2–15 pM*

 Time-independence across the entire binding curve? Y ☑ N ☐

☑ 1.1. *Alternative approach*: measure $k_{off}$.

 $k_{off}$: (0.014 ± 0.003) s$^{-1}$ Calculated equilibration time (5 half-lives): 4 min

☑ 2. Vary the concentration of both binding partners.

 Concentration range of 'trace' binding partner: 2–18 pM (≤15–140 pM)**

 $K_D^{app}$ independent of trace binder concentration? Y ☑ N ☐

 Concentration range showing invariant $K_D^{app}$: 2–18 pM (≤15–140 pM)**

 Binding equation used: ☑ hyperbolic ☑ quadratic

 Binding curves shown? Y ☑ N ☐

 Systematic deviations from the binding curve? Y ☐ N ☑

 $K_D^{app}$: (120 ± 30) pM *(upper limit if dependent on trace binder concentration)*

---

**B. Recommended:**

☑ 1. Test $K_D$ by an independent approach.

 Alternative approach: kinetics

 $K_D^{app}$ from alternative approach: (130 ± 30) pM

☑ 2. Determine the fraction of active protein by titration.

 $K_D$ corrected for active protein fraction? Y ☑ N ☐

 Fraction of active protein: 75–90%

---

**Comments:** * lower and upper limit

 ** lower limits (upper limits)

**Appendix 4—figure 2.** Example of a completed equilibrium binding checklist based on Puf4/RNA binding at 25°C.

**Equilibrium Binding Checklist**

**Binding partner 1 (BP1):** PUF domain of *S. cerevisiae* Puf4 with C-terminal SNAP tag

**Binding partner 2 (BP2):** [5'-$^{32}$P]-AUGUGUAUAUUAGU RNA

**Method:** native gel shift

**CONDITIONS:**  **Temperature:** 0°C   **Buffer & pH:** 20 mM HEPES-K/Na, pH 7.4

**Salt(s):** 2 mM MgCl$_2$, 100 mM KOAc

**Other:** 2 mM DTT, 0.2% Tween 20, 5% glycerol, 0.1 mg/ml BSA

**A. Required:**

☑ 1. Vary incubation time to test for equilibration.

Time range: 0.5–24 hr    Number of time points: 4

BP1 concentration(s): 0.0036–205 nM    BP2 concentration(s): 2–15 pM*

Time-independence across the entire binding curve?    Y ☐  N ☑**

☑ 1.1. *Alternative approach*: measure $k_{off}$.

$k_{off}$: (2.92 ± 0.17) × 10$^{-5}$ s$^{-1}$    Calculated equilibration time (5 half-lives): 33 hr

☑ 2. Vary the concentration of both binding partners.

Concentration range of 'trace' binding partner: 1–27 pM (≤ 9.5–260 pM)***

$K_D^{app}$ independent of trace binder concentration?    Y ☑  N ☐

Concentration range showing invariant $K_D^{app}$: 1–27 pM (≤ 9.5–260 pM)***

Binding equation used:    ☐ hyperbolic    ☑ quadratic

Binding curves shown?    Y ☑  N ☐

Systematic deviations from the binding curve?    Y ☐  N ☑

$K_D^{app}$: (1.39 ± 0.09) pM    *(upper limit if dependent on trace binder concentration)*

**B. Recommended:**

☑ 1. Test $K_D$ by an independent approach.

Alternative approach: kinetics

$K_D^{app}$ from alternative approach: (1.02 ± 0.08) pM

☑ 2. Determine the fraction of active protein by titration.

$K_D$ corrected for active protein fraction?    Y ☑  N ☐

Fraction of active protein: 75–90%

**Comments:**    * lower and upper limit

** not shown for [P]$_{total}$ ≤ $K_D$ (see A1.1)

*** lower limits (upper limits)

**Appendix 4—figure 3.** Example of a completed equilibrium binding checklist based on Puf4/RNA binding at 0°C.

