## [Decision Letter]

**Acceptance summary:**

Given the ubiquitous nature of binding measurements in the literature, including newly emerging high-throughput approaches, this manuscript addresses an important and timely topic. This manuscript is particularly compelling in providing an easy-to-follow set of practical guidelines exemplified with relevant binding data. The authors' approach to this important topic is highly pedagogical and should be a must-read for anyone with the ambition to quantitatively characterize binding equilibria.

**Decision letter after peer review:**

Thank you for submitting your article "How to measure and evaluate binding affinities" for consideration by *eLife*. Your article has been favorably reviewed by two peer reviewers, one of whom is a member of our Board of Reviewing Editors, and the evaluation has been overseen by John Kuriyan as the Senior Editor. The reviewers have opted to remain anonymous.

The reviewers have discussed the reviews with one another and the Reviewing Editor has drafted this decision to help you prepare a revised submission.

Summary:

In this manuscript, a review of 100 studies reporting on binding measurements is presented, allowing the authors to identify and illustrate a number of pitfalls and issues that often adversely affect the reliability and meaningful biological interpretation of binding equilibrium measurements. Using example binding measurements to illustrate the most relevant points, the authors provide a straightforward, practical set of guidelines in terms of a step-by-step “checklist” that can be followed to ensure acquisition of high-quality data best suited to quantitatively describe simple binding equilibria. Given the ubiquitous nature of binding measurements in the literature, including newly emerging high-throughput approaches, this manuscript addresses an important and timely topic. While there may be at least partial overlap with previously published literature on this topic, this manuscript is particularly compelling in providing an easy-to-follow set of practical guidelines exemplified with relevant binding data. The manuscript is well written and accompanied with a number of high-quality and clear illustrations. As such, the authors' approach to this important topic is highly pedagogical and should be a must-read for anyone with the ambition to quantitatively characterize binding equilibria.

Revisions:

The authors should address the following points to further improve clarity of the manuscript:

1) While the need for the equilibration time control is clear, the requirement for changing the concentration of the second species to probe whether ligand depletion could affect Kd measurements seems to be less universal. If a Kd value, as obtained via binding experiment, is substantially larger than the concentration of the labeled species, it would not be strictly necessary to test for potential ligand depletion. It would be important to take this notion into account in the literature survey so as to indicate those cases where a titration regime could indeed be plausible. It would be useful to provide examples from the literature of Kd values that were underestimated due to ligand depletion. The authors should consider emphasizing the specific conditions where ligand depletion might be overlooked (e.g. high-affinity interactions requiring the use of particularly low concentrations of the labeled binding partner, where concentration uncertainty could play a significant role, and a very low active concentration of the protein). That said, varying the equilibration time is an extremely useful control that should be recommended, but perhaps the authors could be more specific as described above. The authors should also consider emphasizing that the inability to obtain a good fit with a hyperbolic function should be considered a serious warning sign that could indicate insufficient equilibration or ligand depletion. The requirement to rigorously report binding curves and fits could be an important part of a binding data reporting standard. In particular for indirect methods such as EMSAs, binding curves and fits are often omitted.

2) Cpf1 is discussed as an example where affinity is substantially underestimated. However, this particular example appears more complicated and likely requires factors other than insufficient incubation time to be considered: First, in one of the studies reporting a 1000-fold lower affinity, a k_off_ of 1/(several seconds) was directly measured using a smFRET assay; Second, the experimental conditions that are known to affect binding were different in all three studies being compared, including temperature, buffer composition (specifically, divalent ions), as well as RNA and DNA sequences; Moreover, in some cases, the Cpf1 proteins were from different species (Strohkendl et al., 2018, and the study reporting the lowest affinity).

Indeed, this illustrates another common mistake when reporting or using binding affinities: treating them as constant values rather than functions of many variables, and ignoring experimental conditions and other important details when comparing the values. It would be important (and educational) to emphasize this in the manuscript, see also minor points for more details. This all being said, Cpf1 still makes a good case for the authors' main point regarding the need to prove that binding is at equilibrium, since the lack of this proof in the study reporting the lowest affinity creates a lot of confusion.

3) In Appendix 3, Weeks and Crothers, 1992 is cited for a precise competitive binding equation for the case of Kd,comp close to total concentration of P, but the solution to the quadratic equation in Weeks and Crothers does not represent a general equation for competitive binding. Instead, the same approximation as in Lin and Riggs (Kd,comp>>total concentration of P) is assumed and the equation is solved for theta to obtain a binding curve rather than a single point for theta=0.5. This approach should still be considered superior when compared to determining Kd,comp from a single data point since it takes all the other data of the curve into account. However, this approach still cannot be used in case of comparable affinities for competitor and labeled ligand. Instead, a general competitive binding curve should be used that represents a correct solution to the cubic equation (see, for example, PMID:
7875313).

---

## [Author Response]

Revisions:The authors should address the following points to further improve clarity of the manuscript:1) While the need for the equilibration time control is clear, the requirement for changing the concentration of the second species to probe whether ligand depletion could affect Kd measurements seems to be less universal. If a Kd value, as obtained via binding experiment, is substantially larger than the concentration of the labeled species, it would not be strictly necessary to test for potential ligand depletion. It would be important to take this notion into account in the literature survey so as to indicate those cases where a titration regime could indeed be plausible.

We updated our literature survey to distinguish between studies in which titration is and is not plausible, and have included additional analysis of literature data based on the concentration of the limiting species relative to the reported *K*_D_ values (see revised Figure 1 and newly added Figure 1—figure supplement 2, as well as accompanying text changes). While most authors appear to be aware of the need to avoid titration, approx. one fifth of all studies used the hyperbolic equation at concentrations that violate the [P]_total_ ≈ [P]_free_ condition. This included several studies in which the reported *K*_D_ was equal to or even lower than the stated limiting species concentration, putting these studies at high risk for inaccuracies that accompany *K*_D_ values determined in the titration regime. In these and other cases varying the concentration of the limiting species would provide a definitive control for titration (and for other concentration-related artifacts). Further, while it becomes *less* likely that there are titration artifacts when reported *K*_D_ values are much greater than the concentration of the second species, one typically does not know for certain if all of the added material is active or if other possible artifacts, such as contamination with another species, are present. Finally, we emphasize that our primary purpose is to make future measurements reliable, so we have tried to make the case that changes are needed in how measurements are typically or often made, without calling out individual studies.

It would be useful to provide examples from the literature of Kd values that were underestimated due to ligand depletion. The authors should consider emphasizing the specific conditions where ligand depletion might be overlooked (e.g., high-affinity interactions requiring the use of particularly low concentrations of the labeled binding partner, where concentration uncertainty could play a significant role, and a very low active concentration of the protein).

We have added a new “Implications of the titration regime” section, analogous to the “Implications of insufficient equilibration” section, where we address the above points.

That said, varying the equilibration time is an extremely useful control that should be recommended, but perhaps the authors could be more specific as described above. The authors should also consider emphasizing that the inability to obtain a good fit with a hyperbolic function should be considered a serious warning sign that could indicate insufficient equilibration or ligand depletion. The requirement to rigorously report binding curves and fits could be an important part of a binding data reporting standard. In particular for indirect methods such as EMSAs, binding curves and fits are often omitted.

We agree that poor fits present an important red flag and now discuss this point in both the equilibration and titration sections. We also amended the checklist (Appendix 4) to include questions about systematic deviations from the fit and whether binding curves are displayed. Nevertheless, it is important not to rely on the quality of fits alone (given potential noise, often sparse data points, subtle and difficult to recognize systematic deviations, or real differences from simple 1:1 binding model). In addition, it is not uncommon for authors to vary the Hill coefficient to compensate for poor fits or to show gel images and derived *K*_1/2_ values without plots or fits of the data. We chose not to elaborate on these practices in order to focus on guidelines for correct measurements rather than on criticisms of unreliable ones.

2) Cpf1 is discussed as an example where affinity is substantially underestimated. However, this particular example appears more complicated and likely requires factors other than insufficient incubation time to be considered: First, in one of the studies reporting a 1000-fold lower affinity, a k_off_ of 1/(several seconds) was directly measured using a smFRET assay;

We have carefully reviewed the paper in question and have consulted with an expert in Cpf1 kinetics. While we recognize the potential for additional complexities, the data strongly support a dominant role of insufficient equilibration in underestimating the affinity. To respond to the first point, the kinetics data in PMID: 29735714 that show rapid dissociation on the scale of seconds refer to targets with 9 mismatches (Figure 3E) or 16 mismatches (Figure S3E). For the fully complementary target, Figure 3A indicates virtually no dissociation after >1 h, consistent with extremely slow equilibration. Figure 3A also suggests slow equilibration (>1 h) for targets with up to 7 mismatches.

Second, the experimental conditions that are known to affect binding were different in all three studies being compared, including temperature, buffer composition (specifically, divalent ions), as well as RNA and DNA sequences; Moreover, in some cases, the Cpf1 proteins were from different species (Strohkendl et al., 2018 and the study reporting the lowest affinity).

We recognize that affinity can vary dramatically across conditions and now address this general point in the manuscript. Nevertheless, in this particular case the conditions used to study *Acidaminococcus* sp Cspf1 were near-identical between PMID: 29735714 and PMID: 30078724, despite a 1000-fold difference between reported *K*_D_ values. We now state “for the same enzyme at similar conditions” in the text and we include details of the conditions below. Moreover, the consistency between the kinetic data in both studies (see the previous point) argues against major acceleration of equilibration in response to the small differences in conditions.

We now only mention the 1000-fold difference between the affinities reported for the same Cpf1 ortholog (AsCpf1) and do not mention the 100,000-fold difference from a study that investigated a different Cpf1 ortholog.

Indeed, this illustrates another common mistake when reporting or using binding affinities: treating them as constant values rather than functions of many variables, and ignoring experimental conditions and other important details when comparing the values. It would be important (and educational) to emphasize this in the manuscript, see also minor points for more details.

We completely agree that the dependence of affinity on specific conditions is an important point and now discuss it in the manuscript.

This all being said, Cpf1 still makes a good case for the authors' main point regarding the need to prove that binding is at equilibrium, since the lack of this proof in the study reporting the lowest affinity creates a lot of confusion.

We agree.

3) In Appendix 3, Weeks and Crothers, 1992 is cited for a precise competitive binding equation for the case of Kd,comp close to total concentration of P, but the solution to the quadratic equation in Weeks and Crothers does not represent a general equation for competitive binding. Instead, the same approximation as in Lin and Riggs (Kd,comp>>total concentration of P) is assumed and the equation is solved for theta to obtain a binding curve rather than a single point for theta=0.5. This approach should still be considered superior when compared to determining Kd,comp from a single data point since it takes all the other data of the curve into account. However, this approach still cannot be used in case of comparable affinities for competitor and labeled ligand. Instead, a general competitive binding curve should be used that represents a correct solution to the cubic equation (see, for example, PMID: 7875313).

We are grateful to the reviewers for pointing us to the general formulation of the competitive binding equation. We have updated Appendix 3 with the new equation and we have added the reference to the cubic equation.